# Design and execution of a verification, validation, and uncertainty quantification plan for a numerical model of left ventricular flow after LVAD implantation

**Alfonso Santiago**[1,2], **Constantine Butakoff**[2], **Beatriz Eguzkitza**[1], **Richard A. Gray**[3], **Karen May-Newman**[4], **Pras Pathmanathan**[3], **Vi Vu**[4], **Mariano Vázquez**[1,2]*

**1** Barcelona Supercomputing Center (BSC), Barcelona, Spain, **2** ELEM biotech, Barcelona, Spain, **3** US Food and Drug Administration (FDA), Silver Spring, Maryland, United States of America, **4** Department of Mechanical Engineering, San Diego State University (SDSU), San Diego, California, United States of America

* mariano.vazquez@bsc.es

## Abstract

### Background

Left ventricular assist devices (LVADs) are implantable pumps that act as a life support therapy for patients with severe heart failure. Despite improving the survival rate, LVAD therapy can carry major complications. Particularly, the flow distortion introduced by the LVAD in the left ventricle (LV) may induce thrombus formation. While previous works have used numerical models to study the impact of multiple variables in the intra-LV stagnation regions, a comprehensive validation analysis has never been executed. The main goal of this work is to present a model of the LV-LVAD system and to design and follow a verification, validation and uncertainty quantification (VVUQ) plan based on the ASME V&V40 and V&V20 standards to ensure credible predictions.

### Methods

The experiment used to validate the simulation is the SDSU cardiac simulator, a bench mock-up of the cardiovascular system that allows mimicking multiple operation conditions for the heart-LVAD system. The numerical model is based on Alya, the BSC's in-house platform for numerical modelling. Alya solves the Navier-Stokes equation with an Arbitrary Lagrangian-Eulerian (ALE) formulation in a deformable ventricle and includes pressure-driven valves, a 0D Windkessel model for the arterial output and a LVAD boundary condition modeled through a dynamic pressure-flow performance curve. The designed VVUQ plan involves: *(a)* a risk analysis and the associated credibility goals; *(b)* a verification stage to ensure correctness in the numerical solution procedure; *(c)* a sensitivity analysis to quantify the impact of the inputs on the four quantities of interest (QoIs) (average aortic root flow $Q_{Ao}^{avg}$, maximum aortic root flow $Q_{Ao}^{max}$, average LVAD flow $Q_{VAD}^{avg}$, and maximum LVAD flow $Q_{VAD}^{max}$); *(d)* an uncertainty quantification using six validation experiments that include extreme operating conditions.

**Funding:** This project was funded in part by the FDA Critical Path Initiative and by an appointment to the Research Participation Program at the

Division of Biomedical Physics, Office of Science and Engineering Laboratories, Center for Devices and Radiological Health, U.S. Food and Drug Administration, administered by the Oak Ridge Institute for Science, and Education through an interagency agreement between the U.S. Department of Energy and FDA to RAG. MV and AS acknowledge the funding from the project CompBioMed2 (H2020-EU.1.4.1.3. Grant number: 823712), SilicoFCM (H2020-EU.3.1.5. Grant number: 777204), and NEOTEC 2019 - "Generador de Corazones Virtuales" ("Ministerio de Economía y competititvidad", EXP - 00123159 / SNEO-20191113). AS salary is partially funded by the "Ministerio de Economía y competititvidad" under the Torres Quevedo Program (grant number: PTQ2019-010528). CB salary is partially funded by the Torres Quevedo Program (grant number: PTQ2018-010290). The funders had no role in study design, data collection and analysis, decision to publish, or preparation of the manuscript.

**Competing interests:** I have read the journal's policy and the authors of this manuscript have the following competing interests: AS, MV, BE and CB have acted as consultants for Medtronic PLC related to Medtronic's HVAD. KMN and VV have an ongoing research study for Abbott, Inc for work on Abbott's HeartMate III. RG and PP need to include the following statement: "The mention of commercial products, their sources, or their use in connection with material reported herein is not to be construed as either an actual or implied endorsement of such products by the U.S. Department of Health and Human Services".

## Results

Numerical code verification tests ensured correctness of the solution procedure and numerical calculation verification showed a grid convergence index $(GCI)^{95\%}$ <3.3%. The total Sobol indices obtained during the sensitivity analysis demonstrated that the ejection fraction, the heart rate, and the pump performance curve coefficients are the most impactful inputs for the analysed QoIs. The Minkowski norm is used as validation metric for the uncertainty quantification. It shows that the midpoint cases have more accurate results when compared to the extreme cases. The total computational cost of the simulations was above 100 [core-years] executed in around three weeks time span in Marenostrum IV supercomputer.

## Conclusions

This work details a novel numerical model for the LV-LVAD system, that is supported by the design and execution of a VVUQ plan created following recognised international standards. We present a methodology demonstrating that stringent VVUQ according to ASME standards is feasible but computationally expensive.

## Author summary

During the regulatory evaluation of newly developed medical devices, the manufacturer provides proof of the device's safety and effectiveness. Historically, the regulatory entities have accepted bench experiments, animal experiments, and human trials as sources of evidence. But as the research questions become more sophisticated, it is becoming increasingly difficult to find trustworthy answers with the classical approach. Numerical modelling opens a new door for the regulatory process with the promise of tackling these new complex questions. But simulations suffer from a fundamental disconnect with practical applications. While most simulations are deterministic, engineering applications have many sources of uncertainty. Furthermore, the numerical model itself can introduce large uncertainties due to the assumptions and the numerical approximations employed. Without forthrightly estimating the total uncertainty in a prediction, decision makers will be ill advised. Recently published standards such as the ASME V&V40 provides a structured manner to provide credibility evidence for the simulation results. This credibility evidence is supported by a thorough check of the numerical model implementation and a quantitative comparison with a physical experiment. This manuscript shows an end-to-end example for the design and execution of a verification, validation and uncertainty quantification (VVUQ) following the ASME V&V40 standard.

## Introduction

Over 5 million people suffer heart failure (HF) in the U.S. alone, with ∼1 million new cases diagnosed annually [1]. Heart transplant is the recommended treatment for the 10% of these patients in Stage D [2] condition. Despite this, there are only 2000 organs yearly available for transplant [3], sufficient for only 0.4% of these patients. The limited organ availability is making left ventricular assist device (LVAD) therapy a leading treatment for the remaining 99.6% patients, with a ∼90% chance of 1-year survival [4].

Mortality and HF status following LVAD implantation are primarily associated with inefficient unloading of the left ventricle and persistence of right ventricular dysfunction [5], and stroke [6]. Optimization of LVAD speed is routinely performed for patients post-implant with a ramp study. Transthoracic echocardiography measurements of cardiac geometry and function are made while LVAD speed is slowly increased over a wide range [7]. The final pump speed is selected by balancing overall cardiac output, efficiency of left ventricle (LV) unloading, and preserving flow pulsatility [5]. Several variables are assessed from standard ultrasound views including LV end-diastolic dimension, LV end-systolic diameter, frequency of Aortic Valve (AoV) opening, degree of valve regurgitation, right ventricle (RV) systolic pressure, blood pressure, and heart rate (HR) at each speed setting. In addition, LVAD pump power, pulsatility index and flow are recorded. For the Thoratec HeartMate II, the ramp speed protocol starts at a speed of $8k[rpm]$ and increases by $400[rpm]$ increments every 2 minutes until a speed of $12k[rpm]$ is reached. As LVAD speed is increased, the LV volume decreases, as does the frequency of AoV opening and flow pulsatility. Excessive LV unloading at higher LVAD speeds increases the demand on the right heart, causing tricuspid regurgitation and also possibly producing suction events, which disrupt the flow into the LVAD inflow cannula.

The clinical practice for LVAD speed selection first ensures that the hemodynamics are compatible with life, *e.g.* a mean arterial pressure greater than 65 mmHg [7] and a minimum cardiac index of $3.6\times10^{-6}$[ms] (2.2[L/min/m$^2$]) of body surface area (BSA) [5] To optimize LV unloading, the interventricular septum position should not bow towards either the left or right. If these conditions are met, the LVAD speed is selected that achieves intermittent AoV opening while maintaining no more than mild mitral regurgitation or aortic insufficiency [8]. *De novo* AoV insufficiency development in LVAD patients is linked to lack of AoV opening. Interestingly, AoV insufficiency occurred in the majority of LVAD patients (66%) whose AoVs remained closed during support, but rarely (8%) in those whose AoVs opened regularly [7].

A patient specific LVAD speed calibration is important for ensuring appropriate cardiovascular support and minimizing the frequency of adverse events related to long-term support. However, the ramp echo study is not performed routinely after the first month post-implant, due to the unjustified expense and inconvenience. A computational tool that predicts cardiac output and aortic valve opening for the subject's characteristics could reduce the ramp study requirements as well as contribute to speed adjustments required over a long-term, even supporting a speed adjustment paradigm that contributes to recovery. As LVADs are a life support therapy that carry a high risk for the patient, the Food and Drug Administration (FDA) rank them in the most exhaustive level of control (Class III) before approving their commercialisation. Therefore, using computational models to predict the behaviour of these devices or to guide design decisions should be accompanied with a stringent validation process to ensure credible results. Validating computational models is a whole challenge by itself, but recent published guidelines tackle this problem.

While scientific computing has undergone extraordinary increases in sophistication, a fundamental disconnect exists between simulations and practical applications. While most simulations are deterministic, engineering applications have many sources of uncertainty arising from a number of sources such as subject variability, initial conditions or system surroundings. Furthermore, the numerical model itself can introduce large uncertainties due to the assumptions and the numerical approximations employed [9]. Without forthrightly estimating the total uncertainty in a prediction, decision makers will be ill advised. To address this issue, multiple standards for industrial guidance has been published like the American Society of Mechanical Engineers (ASME) V&V codes [10–12].

Extensive modeling studies dealing with multiple LVAD factors like ventricular size [13], cannula implantation position [14], implantation depth [15–17] or angulation [18] exist but none of them provide credibility evidence as suggested in the recent ASME V&V40 [11], nor are any guided by AMSE V&V20 [12] which was developed over 10 years ago for demonstrating credibility of computational fluid dynamics (CFD) models. The reason for this is that such a validation requires a thorough comparison of the simulation results against bench or animal experiment measurements and hundreds of executions of the numerical model, which involves a large computational cost.

To our knowledge, this is the first paper describing such a comprehensive computational LVAD setup and using ASME verification, validation and uncertainty quantification (VVUQ) standards to design and execute a VVUQ plan. While our final goal is to predict intra-LV stagnation biomarkers, this manuscript is focused on the credibility assessment of the numerical model. The contributions of this manuscript combines:

1. A deformable ventricle numerical model using a unidirectional fluid-structure interaction (FSI) and 0D aortic impedance model (as in [16]).

2. A novel pressure-driven valve model for mitral and aortic valves.

3. A dynamic pressure-flow (also called H-Q) performance curve for the LVAD boundary condition.

4. A VVUQ plan designed and executed following the ASME V&V40 and V&V20 standards [11, 12].

5. A set of validation metrics to quantify the differences between the simulation and the experiment and that describe the aortic valve flow and total cardiac output.

## Note on the nomenclature

The words "model" and "experiment" may work for any simpler representation of a more complex system (*e.g.* animal, bench or numerical model/experiment). For the sake of brevity, we refer to the bench model/experiment simply as "experiment" and to the numerical model/experiment as "simulation", except if stated otherwise.

## Methods

### Description of the benchtop model

The experiments were performed with the San Diego State University (SDSU) cardiac simulator (CS), shown in Fig 1. This CS is a mock circulation loop of the heart and the circulatory system with an apically implanted LVAD (Abbott HeartMate II) that has been reported previously in [19, 20]. It involves a transparent model of the dilated LV based on an idealised geometry, immersed in a water-filled tank and connected to an external circulatory loop mimicking the systemic circulation. The tank is fully watertight, so when the piston pump generates negative pressure, the LV expands to the end diastolic volume (EDV). The LV used is manufactured from platinum-cured silicone rubber (Young's modulus $E = 6.2 \times 10^5$[Pa] at 100% elongation and ultimate tensile strength of $P_{max} = 5.52 \times 10^6$[Pa] at 400% elongation). Porcine valves were used in both the aortic position (26[mm] Medtronic 305 Cinch) and the mitral position (25[mm] 6625 Carpenter Edwards Bioprosthesis). Tygon tubing (16[mm] diameter) replaced the HeartMate II outflow graft and was connected to the ascending aorta at a 90[°] angle approximately 15[mm] distal to the aortic root. The circulating fluid was a viscosity-matched blood

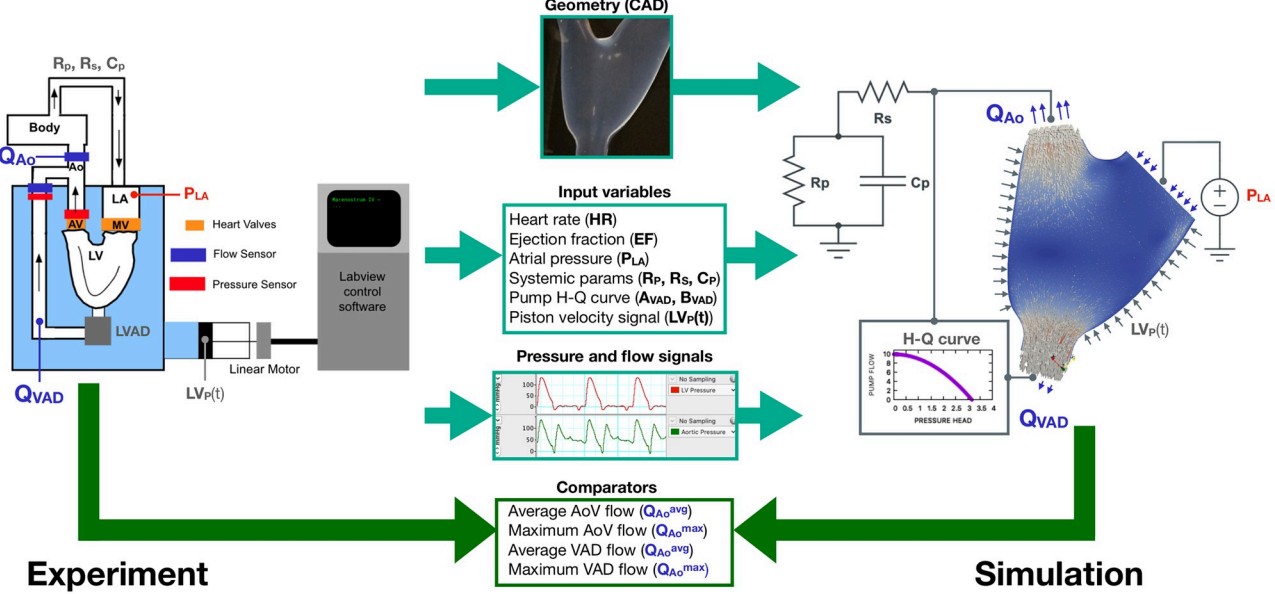

**Fig 1. Leftmost side: Schematic of the experiment setup.** LA: left atrium, MV: Mitral valve, AV: Aortic valve, Ao: aorta. Flow and pressure sensors are indicated in blue and red respectively. The body lumped system is afterwards characterised with a three element Windkessel model with parameters $R_p$, $R_s$ and $C_p$. Center: variables extracted from the benchtop experiment to create the simulation and address the uncertainty quantification (UQ). Rightmost side: Schematic of the simulation, including the lumped models for the boundary conditions. The measured $P_{LA}$ is imposed in the mitral valve. The measured $R_p$, $R_s$ and $C_p$ are used in a three element Windkessel boundary condition at the Aortic valve output. The H-Q curve retrieved and measured is used as a dynamic boundary condition in the LVAD outflow. The benchtop piston dynamics is used as boundary condition for the deformable LV geometry.

analogue consisting of 40[%] glycerol (viscosity of $\mu = 3.72 \times 10^{-3}$[Pas$^{-1}$] at 20[$^\circ C$]) and saline [21]. Constant mitral pressure is achieved by using a large open reservoir for the left atria (LA), maintaining the LA pressure constant thorough the studies. A fluid circuit composed of partially clamped tubing and compliance chambers is used to physically represent and tune the systemic circulation. This circuit can be mathematically represented by a 3-element Windkessel model with $R_p^{Ao} = 1.7 \times 10^8$[Pa $\cdot$ s/m$^3$], $C_p^{Ao} = 1.2 \times 10^{-8}$[m$^3$/Pa], $R_s^{Ao} = 7.8 \times 10^6$[Pa $\cdot$ s/m$^3$], following the method in [22]. This lumped representation of the circulatory system allows characterising the outlet boundary condition in the numerical model.

During diastole the silicone ventricle dilates allowing the filling with fluid from the atrial chamber. During systole the ventricle is compressed expelling fluid through the aortic valve. Through this process the LVAD is continuously extracting fluid out of the LV. If the LVAD speed is high enough, the aortic valve remains hemodynamically closed during systole.

The total aortic flowrate $Q_{TAo}$ is measured with a Transonic TS410 20PXL clamp-on flow meter (resolution: $8.3 \times 10^{-7}$[m$^3$/s] (50[$ml/min$]), maximum zero offset: $5 \times 10^{-9}$[m$^3$/s] (0.3[$ml/min$]), and absolute accuracy: 10[%]). The LVAD flowrate $Q_{VAD}$ is measured with a Transonic TS410 10PXL flow meter (resolution: $1.6 \times 10^{-7}$[m$^3$/s] (10[$ml/min$]), maximum zero offset: $1.0 \times 10^{-9}$[m$^3$/s] (0.06[$ml/min$]), absolute accuracy: 10[%]). The aortic valve flow $Q_{Ao}$ is calculated as $Q_{AoV} = Q_{TAo} - Q_{VAD}$. Pressure in the LV and the aortic root are measured with two Icumed TranspacIV (sensitivity: 666.65[Pa](5[mmHg]) $\pm$1%, zero offset: 3333.3[Pa] (25 [mmHg])). Signals are amplified collected with a ADinstruments Powerlab DAQ device (sampling frequency: 200[Hz], impedance 1[$M\Omega$]@1[$pF$], resolution: 16[bit]) and processed with ADInstruments LabChart version: 1.8.7. [23].

Two beating modes and three pump speeds are used for six validation experiments (described ahead in section *Validation points and ranges*). The beating mode 22[%]@68.42 [*bpm*] has *EF* = 22[%] and *HR* = 68.42[*bpm*] with end systolic volume (ESV) = 180.0×10$^{-6}$[$m^3$] (180[$cm^3$]) and EDV = 230.0×10$^{-6}$[m$^3$] (230[cm$^3$]). The beating mode 17[%]@61.18[*bpm*] has *EF* = 17[%] and *HR* = 61.18[*bpm*] with ESV = 180.0×10$^{-6}$[m$^3$] (180[cm$^3$]) and EDV = 216.86×10$^{-6}$[m$^3$] (216.86[cm$^3$]). The Ejection Fraction (EF), the EDV, and the ESV are linked by the EF equation:

$$EF = \frac{EDV - ESV}{EDV}.$$

As the ESV is fixed by the silicone ventricle volume, there is a direct correlation between EF and EDV. From now on, we only characterise the case as a function of the EF. Both beating modes correspond to a New York heart association (NYHA) Class IV HF patient [24]. The pump speeds used for the validation points are 0[*rpm*], 8k[*rpm*] and 11k[*rpm*]. To avoid back-flow, the LVAD outflow conduit were clamped in the 0[*rpm*] experiments so the system mimics a pre-LVAD baseline rather than a heart implanted with a LVAD turned off. The pressure-flow (also called H-Q) performance curves are experimentally retrieved for each pump speed (see S4 Data) and used afterwards as input for the simulation. Approximating a pump pressure-flow curve with a quadratic fit is a common engineering practice.

The SDSU-CS has been widely used in academia and industry, and its reproducibility addressed, not only for LVADS [19, 25–30], but also valves [23, 31, 32], and with combined devices [20, 33]. In this work use retrospective experimental data for the validation with UQ.

### Description of the numerical model

**Overall simulation pipeline.**   The computational domain is created from the exact same computer geometry used to manufacture the silicone ventricle (refer to Fig 1). The FSI problem requires the construction of two meshes. The solid mechanics mesh is created directly from the original geometry, containing at least four linear tetrahedra in the wall thickness, obtaining a total of 200k elements. The CFD mesh was created by closing the solid domain and extruding the inlets and outlets to ensure flow development. The geometry is discretized including a boundary layer valid for *Re* < 4000[−] [34], using linear tetrahedra, pyramids and pentagons. The final mesh has 1.6M elements. For the time discretisation, a first order trapezoidal rule with a time step of 0.00428[s] was used in every case, which matches the experimental setup sampling period.

The FSI problem can be tackled by a unidirectional or a bidirectional approach [35]. In the former approach, the solid problem unilaterally deforms the fluid mesh. In the latter approach, an iterative process is requires to balance the internal forces of the solid problem with the surface pressure of the fluid problem. To obtain a computationally inexpensive and accurate way of deforming the ventricle, a unidirectional FSI approach is used to deform the LV. This same approach is used in [16], where a pressure is imposed in the external solid domain which afterwards deforms the CFD domain between the ESV and the EDV. In this unidirectional FSI approach the solid domain is exclusively used to impose the boundary deformation and velocity in the fluid domain, but no force is fed back to the solid problem, as it would be in an iterative bidirectional FSI formulation [35]. We justify this choice from the working principle of the experimental set-up (section *Description of the benchtop model*). The piston forces volume changes in the silicone ventricle independently of the internal ventricle pressure.

Once the simulation pipeline is completed, the input files are modified to work as a template. This template is used by Dakota server (DARE) (described in section DARE) for the sensitivity analysis (SA) and the UQ analysis.

**Description of the solver.**   Here we briefly describe the numerical model used, highlighting only the novel components. The incompressible, Newtonian fluid is modelled by the Navier-Stokes equations in Alya, the Barcelona Supercomputing Center (BSC) in-house tool for simulations [36]. The Navier-Stokes equations are solved using an an Arbitrary Lagrangian-Eulerian (ALE) formulation, allowing the fluid domain to deform:

$$\rho \frac{\partial v_i}{\partial t} + \rho \left(v_j - v_j^d\right)\frac{\partial v_i}{\partial x_j} + \frac{\partial}{\partial x_j}\left[+p\delta_{ij} - \mu\left(\frac{\partial v_i}{\partial x_j} + \frac{\partial v_j}{\partial x_i}\right)\right] = \rho f_i \qquad \text{and} \qquad \frac{\partial v_i}{\partial x_i} = 0, \quad (1)$$

where $\mu$ is the dynamic viscosity of the fluid, $\rho$ the density, $v_i$ the velocity, $p$ is the mechanical pressure, $f_i$ the volumetric force term and $v_j^d$ is the domain velocity. As the fluid domain deforms due to the imposed boundary displacements, the deformation for the inner nodes have to be computed. For this, we use the technique proposed in [37]. Thenumerical model is based on the finite elements method (FEM), using the algebraic subgrid scale (ASGS) as in [38] for stabilisation. In order to solve this system efficiently in supercomputers, a split approach is used [39]. The Schur complement is obtained and solved with an Orthomin(1) algorithm [40] with weak Uzawa operator preconditioner. The momentum equation is solved twice using generalized minimal residual method (GMRES) with Krylov dimension 100 and a diagonal preconditioner. For the continuity equation a deflated conjugate gradient algorithm is used. Solid mechanics is modelled via linear momentum balance [41], using a neo-Hookian formulation to represent the Platinum-cured silicone [42]. As described in section section *Description of the benchtop model*, the solid material bulk modulus is $K = 1\times10^6$[Pa] and its shear modulus is $G = 2\times10^5$[Pa] (corresponding to a Poisson ratio of $v = 0.4$[−]). The implicit formulation is solved with a using a total Lagrangian formulation and the GMRES algorithm for linear systems' where Newton method is the nonlinear solver.

A complete description of the fluid- electro- mechanical model can be found in [43]. The referenced work shows the governing equations and the solution strategy for cardiac electrophysiology, mechanical deformation, and the ventricular hemodynamics. The cited manuscript also describes the strategy for both bidirectional couplings present in the three physics problem.

**Initial and boundary conditions.**   The Mitral inlet has a constant pressure of $P_{LA}$. The aortic model has a 0D Windkessel model with three components: a resistor $R_s$, serially connected with an $RC$ parallel impedance $R_p$, and $C_p$ [22]. On the deforming ventricular walls, as well as on the rest of the fluid domain, the velocity the domain deformation is imposed. This is $v_i|_{\Gamma_d} = \partial b_i/\partial t$, where $v_i|_{\Gamma_d}$ is the velocity at the deformed boundary. The LVAD outlet has a specific type of boundary condition that will be thoroughly described in section *LVAD boundary condition*. the rest of the boundaries have $v_i|_{\Gamma_r} = 0$[m/s]. As for the initial condition in the unknowns, the initial fluid velocity is $v_i|_{t=0} = 0$[m/s].

**Valve modelling.**   Mitral and aortic valves are modelled through a pressure-driven porous layer in the valvular region. This porous media add an isotropic force to the right-hand side of the momentum equation with the shape $f_i^P = \sigma_{ij}^P v_j$, where $\sigma_{ij}^P = PI_{ij}$ where $P$ is the material porosity and $I_{ij}$ the identity matrix. This strategy provides a robust numerical scheme against the potential ill-conditioned stage of confined fluid. To ensure a smooth change with potentially abrupt changes in the transvalvular pressure, the porosity is driven through a hyperbolic

tangent as:

$$P = P_{max}\left[1 + \tanh\left(\frac{\Delta p^V - \Delta p_{ref}^V}{s}\right)\right], \tag{2}$$

where $P_{max}$ the maximum possible porosity, $s$ the slope of the curve, $\Delta p^V$ the transvalvular pressure drop, and $\Delta p_{ref}^V$ a reference pressure gradient. In practice if $\Delta p^V \gg \Delta p_{ref}^V$ the valve is closed and if $\Delta p^V \ll \Delta p_{ref}^V$ the valve is open. To avoid spurious valve opening and/or closing due to transient peaks in the tranvalvular pressure gradients, the measures are filtered using a median filter.

**LVAD boundary condition.** Pump performance can be characterised with pressure-flow curves for each speed of the pump's rotor. These pressure-flow curves (also called H-Q curves) provide a relation between the pressure difference between the pump inlet and outlet, and the flow the pump can provide at that speed.

Let $\Delta p^{VAD} = P_{Ao} - P_{LV}$ be the pressure difference between the outlet and the inlet of the pump and $Q_{VAD}$ the flow through the pump. The pressure-flow relationship can be approximated with a quadratic equation as:

$$\Delta p^{VAD} = A_{VAD} + B_{VAD}Q_{VAD} + C_{VAD}Q_{VAD}^2 \tag{3}$$

For each $\Delta p^{VAD}$ there is a single $Q_{VAD}$ and vice versa. This relation can be used as a boundary condition, imposing a flowrate for a calculated pressure difference $\Delta p^{VAD}$. With this, the LVAD boundary condition keeps a flowrate constrained with Eq 3. The method to calculate the variable ranges for the pump model inputs is explained in the S4 Data. The fitting coefficients with their errors are shown in Table 1.

**DARE.** Executing the SA and the UQ during the VVUQ process requires generating thousands of inputs for the simulation code, submitting the jobs, processing the simulation results, and extracting the quantities of interest (QoIs) out of the physical field results. For this, we created DARE.

DARE is an automating tool that works coupled with Sandia's Dakota (version 6.12) [44, 45] and allows automatically encoding, submitting and retrieving jobs to any high performance computing (HPC) infrastructure (Fig 2). Dakota allows to characterise and sample model inputs for multiple analysis types like SA, UQ or optimisation. The Dakota+DARE pair runs in a computer external to the HPC machine for as long as the analysis under execution may last (up to several weeks in this work). DARE receives Dakota's chosen inputs, processes the simulation templates with an encoder, submits the job to the supercomputer queue, waits for the jobs to be finished and processes the final results to feed Dakota with the obtained outputs. A combination of Dakota's restart capabilities with DARE failure capture capabilities makes this a robust framework for the required analysis.

**Table 1. Fitting coefficients ($a_{VAD}$, $b_{VAD}$, and $c_{VAD}$) and fitting errors $\epsilon$ of the H-Q performance curves.** The pump speed is measured in [$rpm$]. The units are $a_{VAD}$[Pa], $b_{VAD}$[Pa · s/m$^3$], and $c_{VAD}$[Pa · s$^2$/m$^6$].

| Pump speed | $a_{VAD} \pm \epsilon$ / $b_{VAD} \pm \epsilon$/$c_{VAD}$ |
| --- | --- |
| 0k | $0.0 \pm 0.0$ / $0.0 \pm 0.0$ / $0.0$ |
| 8k | $1.17 \times 10^4 \pm 1.44 \times 10^2$ / $-7.72 \times 10^7 \pm 2.85 \times 10^6$ / $0.0$ |
| 11k | $2.17 \times 10^4 \pm 4.89 \times 10^2$ / $-9.02 \times 10^7 \pm 6.15 \times 10^6$ / $0.0$ |

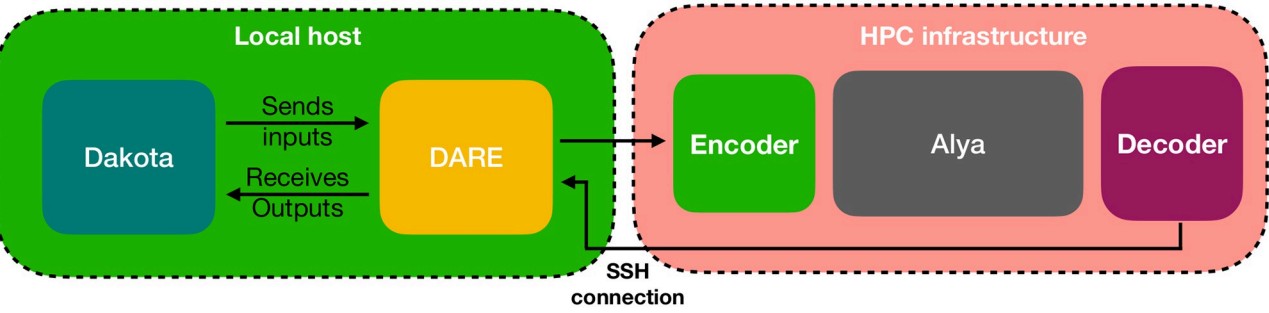

**Fig 2. Scheme of DARE building blocks.**

### Design of the VVUQ plan through V&V40 risk-based credibility assessment

We performed credibility assessment by following the ASME V&V 40 [11] standard. The standard provides a framework for assessing the relevance and adequacy of the completed VVUQ activities for medical devices. Applying the standard requires a set of preliminary steps to determine the required level of credibility for the model. These preliminary steps are to identify: *(1)* the question of interest, this is the question the tool will find an answer to; *(2)* the context of use (CoU), this is the specific role and scope of the computational model; *(3)* the QoI, these are the simulation outputs relevant for the CoU; *(4)* the model influence, this is the contribution of the model in making a decision; *(5)* the decision consequence, this is the possibility that incorrect model results might lead to patient harm; and *(6)* model risk, which is based on model influence and decision consequence. Once these items are identified, goals for the credibility evidence can be defined and the VVUQ plan designed.

**Question of Interest.** For an apically implanted LVAD, does the selected pump speed produce: *(a)* complete aortic valve opening ($Q_{Ao} > 5 \times 10^{-6}$[m³/s] (0.3[L/min])); and *(b)* a Cardiac output compatible with life ($Q_{Ao}^{avg} + Q_{LVAD}^{avg} > 7 \times 10^{-5}$[m³/s] (4.2[L/min])) for a range of HR and EF covering a HF patient population?

**Context of Use (CoU).** The heart-LVAD computational model may be used by design engineers to assist in the preclinical development of LVAD, by characterising aortic root, LVAD and intra-LV flows for a given pump speed. The goal of the heart-LVAD computational model is to provide a computational replica of a benchtop experiment for a quantitative analyses in parametric explorations. The heart-LVAD computational model by no means is replacing animal experiments or clinical trials, but augmenting the totality of evidence.

**Quantities of Interest (QoI).** These are the simulation outputs relevant to the CoU. For completeness, here we repeat that the QoIs are the maximum and average flows through the outlet boundaries (LVAD flow $Q_{VAD}$ and aortic root flow $Q_{Ao}$).

**Model influence.** Although the numerical test will augment the evidence provided by the bench test to aid design, they do not qualify the safeness of the device. This meaning animal testing and clinical trials are still required during the regulatory submission to prove safety and efficacy of the device. Therefore the computational model influence can be categorised as low (see Table 2).

**Decision consequence.** While the CoU specifies the usage of the model for design iterations, the results could be used to make indirect decisions that affect the patients' health. If the model fails to make accurate predictions for the question of interest, could advice for an operating condition that produce either: *(a)* low cardiac output or *b* a permanently closed aortic

**Table 2. Risk map.** Adapted from [46].

| Model influence | high | 3 | 4 | 5 |
|---|---|---|---|---|
| | med | 2 | 3 | 4 |
| | low | 1 | 2 | 3 |
| | | low | med | high |
| | | | Decision consequence | |

valve. These might lead to thromboembolic events, aortic regurgitation or death. Therefore the decision consequence is categorised as high (see Table 2).

**Risk assessment.** As the model influence has been categorised as "low" and the decision consequence as "high", the LV-LVAD model is categorised with a risk of 3 on the 1–5 scale from Table 2, therefore requiring a medium level goals in the VVUQ plan.

**Translation of model risks into credibility goals.** The ASME V&V40 standard [11] defines 13 credibility factors (some containing sub-factors) that break down the assessment of the VVUQ activities. Once the risk associated with the modelling tool has been determined, the next stage of the V&V40 pipeline is defining a ranking (gradation) for each factor sorted by increasing level of investigation, and then selecting a credibility goal for each factor based.

Table 3 lists the 13 credibility factors and sub-factors. Gradations for each factor are provided in the S1 Data. For most credibility factors, the gradation proposed in the V&V40 standard is used. Table 3 summarizes the maximum possible score in the gradation, the targeted goal and the achieved score. The targeted goal also includes the description required to achieve that score. Per V&V40, goals were chosen so that model credibility is generally commensurate with model risk. Therefore, for most factors, a medium level or higher goal was chosen. The rationale behind the chosen goals is provided in the S2 Data.

## Design and goals of the VVUQ plan

This section explains the VVUQ activities carried out to achieve the credibility goals defined. The VVUQ plan has been designed following [9, 11, 12].

**Steps of the VVUQ plan.**

1. **Provide verification evidence:** SQA practices should be followed to ensure reproducibility and traceability. Numerical code verification (NCV) is mandatory to ensure correctness in the coding of the models. Numerical calculation verification is mandatory to ensure a sufficient spatial discretisation of the problem.

2. **Execute a sensitivity analysis in the operating range:** A non-linear global SA within the operating range of the cases should be executed to: *(a)* understand the impact of each input on the QoIs, and *(b)* Safely reduce the number of input variables for the UQ through Pearson's ρ and Sobol indices analyses. The goal of step *(b)* has a direct impact in the UQ as it reduces its computational cost.

3. **Perform validation with uncertainty quantification:** The reduced input model obtained from the SA is used to execute the UQ analysis. At least a middle point and the extreme cases of the operation envelope should be investigated. A comparison of the QoIs' distributions is required including a validation metric that allows quantitatively comparing the results between validation points and against other similar works or future projects.

**Table 3. ASME V&V40 credibility factors [11] analysed on the risk-based assessment.** The table shows the maximum possible score ("Max." column), the desired goal ("Goal" column) and the obtained score ("Obt." column). The goal column also includes the description of the activity to achieve that gradation.

| Aspect | | | Evaluation | | |
|---|---|---|---|---|---|
| | | | Max. | Goal | Obt. |
| 1. Verification (Sec. *Verification*) | 1.1. Code | 1.1.1. software quality assurance (SQA) | C | B [SQA procedures are specified and documented.] | C |
| | | 1.1.2. Numerical code verification | D | C [The numerical solution is compared to an exact solution.] | D |
| | 1.2. Calculation | 1.2.1. Discretisation error LightGray | C | B [Convergence analysis are performed obtaining stable behaviours.] | C |
| | | 1.2.2. Numerical solver error | C | B [Solver parameters are based on values from a previously verified model.] | B |
| | | 1.2.3. User error | D | B [Key inputs and outputs were verified by the practitioner.] | C |
| 2. Validation (Secs. *Sensitivity analysis* and *Validation with uncertainty quantification*) | 2.1 Computational model | 2.1.1. Model form | C | B [Influence of some assumptions is explored.] | B |
| | | 2.1.2 Model inputs / 2.1.2.1. Quantification of sensitivities | C | B [A SA of the expected key parameters is performed.] | B |
| | | 2.1.2.2. Quantification of uncertainties | D | B [UQ is executed on expected key inputs but not propagated to the QoIs.] | C |
| | 2.2. Comparator | 2.2.1. Test samples / 2.2.1.1. Quantity of test samples | C | A [A single sample is used.] | A |
| | | 2.2.1.2. Range of characteristics of test samples | D | A [A single test condition is examined.] | A |
| | | 2.2.1.3. Measurements of test samples | C | B [One or more key characteristic are measured.] | C |
| | | 2.2.1.4. Uncertainty of test samples measurements | C | A [Characteristics uncertainty is not addressed.] | A |
| | | 2.2.2. Test conditions / 2.2.2.1. Quantity of test conditions | B | B [Multiple test conditions.] | B |
| | | 2.2.2.2. Range of test conditions | D | B [Test conditions representing a range of conditions near nominal range are examined.] | C |
| | | 2.2.2.3. Measurements of test conditions | C | B [One or more key test conditions are measured.] | B |
| | | 2.2.2.4. Uncertainty of test conditions measurements | C | B [UQ of the test conditions incorporated instrument accuracy only.] | B |
| | 2.3. Assessment | 2.3.1. Equivalence | C | B [The types of all inputs are similar, but ranges are not equivalent.] | C |
| | | 2.3.2. Output Comparison / 2.3.2.1. Quantity | B | B [Multiple outputs were compared.] | B |
| | | 2.3.2.2. Equivalency of output parameters | C | B [Most types of outputs are similar.] | C |
| | | 2.3.2.3. Rigour of output comparison | C | B [Comparison was performed by arithmetic difference.] | C |
| | | 2.3.2.4. Agreement of output comparison | C | B [The level of agreement is satisfactory for some key comparisons.] | B |
| 3. Applicabilty (Sec. *Discussion on the V&V40 credibility factors: Achieved score*) | 3.1. Relevance of the Quantity of interest | | C | B [A subset of the QoIs are identical to those for the CoU.] | C |
| | 3.2. Relevance of the validation activities to the CoU | | D | B [There is partial overlap between the ranges of the validation points and the CoU.] | C |

4. **Adequacy assessment:** Evaluate if the simulation credibility evidence is good enough to safely answer the question of interest.

**Code and calculation verification.**    Code and calculation verification tests provide evidence of the correctness in the translation of the governing equations and numerical solution procedure. It is enclosed by SQA that provides means to monitor the software engineering processes and ensure traceability of the changes. Numerical code verification tests provide a metric of the correctness of the governing equations implementation. These types of tests generally compare the solution obtained by the simulation code with a known analytical solution. Numerical calculation verification is intended to bound the error introduced by the numerical discretisation. It involves the comparison of results in increasingly refined discretisations to estimate the numerical error in the final simulations. For this manuscript, we choose to execute stationary, dynamic, 2D, and 3D numerical code verification tests to ensure the correctness of the governing equations implementation. Finally, increasingly complex numerical calculation verification tests are shown that allow bounding the discretisation error.

**Sensitivity analysis.**    A SA [47] is a statistical tool that allows quantifying the impact of each input variable in each QoI of the model. It is helpful as it allows to rank the input variables based on their contribution to the variation of the model output. On the one hand, identifying the less relevant inputs allows to reduce the dimensionality of the problem, as the less relevant inputs can be safely avoided to decrease the computational cost during UQ. On the other hand, reducing the experimental uncertainty of the most relevant inputs identified is critical to obtain accurate model predictions. The reason for this is that a large uncertainty in a highly impactful input will produce a large uncertainty in the model output.

The SA is carried out by firstly sampling the input values using latin hypercube sampling (LHS) for all the shared input variables. Later, these samples are used to execute independent CFD simulations. In this work we execute two types of SA: *i)* local SA via Pearson coefficients [48], and *ii)* global SA through total Sobol indices calculation [49] by relying on a 5-th order polynomial chaos expansion (PCE) [50]. While having both, Pearson coefficient and Sobol indices may seem redundant, the former is simpler to understand and implement than the latter. This eases the task of reproducing and comparing the manuscript results. Both analyses are performed by relying on 500 samples. Pearson´s coefficient analysis is a first order approach that provides insights of the model behaviour with an accessible and straightforward method. However, it is a measure of the linear association between the inputs and the outputs and it is valid only under Pearson´s assumptions [51] of linear and homoscedastic data with no multivariate outliers. For complex data distributions, Sobol indices are a better fitted method that provides information of the importance of each input taking into account complex factors like nonlinearities, input interactions, and sample dispersion. Sobol´s global SA rely on high-order integrals to accurately calculate the indices. These integrals require a relatively large number of samples to be evaluated. In this manuscript, the original 500 samples are used to fit a PCE emulator which is afterwards used to obtain the 5000 samples required for the Sobol integrals. The PCE polynomial order has been chosen to balance computing time and fitting accuracy. Further description on the Sobol indices and the PCE method can be found in [49, 50].

**Uncertainty quantification.**    Validation involves measuring the difference between both sources of predictions, the experiment and the simulation. These two sources are subject to different types of uncertainties that should be identified as part of a UQ analysis.

The measuring instruments in the experiment introduce the measuring error, while user error is introduced by the experimentalist variability. In the experiment on this manuscript,

**Table 4. Table of model inputs and their uncertainty characterisation.**

| Variable | Symbol | Classification | Characterisation | value range |
|---|---|---|---|---|
| Density | $\rho$ | deterministic | exact value | $1100[\text{kg/m}^2]$ |
| Viscosity | $\mu$ | deterministic | exact value | $0.00372[\text{Pa} \cdot \text{s}]$ |
| Heart rate | $HR$ | aleatory | uniform | $[40, 120][\text{bpm}]$ |
| Atrial pressure | $P_{LA}$ | aleatory | uniform | $[0.5, 1.5] \times 10^3[\text{Pa}]$ |
| Ejection fraction | $EF$ | aleatory | uniform | $[0.1, 0.35][-]$ |
| Aortic serial resistance | $R_s^{Ao}$ | aleatory | uniform | $[5, 20] \times 10^6[\text{Pa} \cdot \text{s/m}^3]$ |
| Aortic parallel resistance | $R_p^{Ao}$ | aleatory | uniform | $[50, 200] \times 10^6[\text{Pa} \cdot \text{s/m}^3]$ |
| Aortic parallel Capacitance | $C_p^{Ao}$ | aleatory | uniform | $[5.0, 20.0] \times 10^{-7}[\text{m}^3/\text{Pa}]$ |
| Constant VAD coefficient | $A_{VAD}$ | aleatory | uniform | $[0.025, 1] \times 10^1[\text{m}^3/\text{s}]$ |
| Linear VAD coefficient | $B_{VAD}$ | aleatory | uniform | $[0.5, 5] \times 10^{-4}[\text{Pa} \cdot \text{s/m}^3]$ |

each execution of the experiment contains multiple beats. Therefore, even for the same set of inputs and due to the measuring instrument error (see section *Description of the benchtop model*), there will be a dispersion in the QoIs that requires the the measurement error to be quantified. As we use retrospective experimental data not specifically thought for VVUQ, there is only a single execution of the experiment for each of the six validation points. This hampers quantification of the user error. To tackle this issue we add a 10% error range in the QoIs measured in the experiment to account for the user error. As there is no other information on that user error shape, no probability distribution can be assumed. The numerical error in the simulation tool is estimated by the code and calculation verification (see S3 Data), and the input error quantified during the UQ. All the parameters shared by both the experimental and numerical models are listed and classified in Table 4.

Each one of the model variables can be characterised as one of the following three: *(a)* deterministic, when their values are known *(b)* aleatory, when the variable is uncertain due to inherent variation and can be characterised with a cumulative distribution function (CDF); *(c)* epistemic, when the variable is affected by reducible uncertainty due to lack of knowledge and it can be represented with a bounded interval. The inputs of the model should be characterised by one of these three categories and treated accordingly. The uncertain variables are sampled using LHS in the uniform ranges identified in Table 4. While SA is executed for all the variables referred in Table 4, the UQ may be executed on a reduced set of inputs. If the SA study concludes there are variables with little to no impact in the QoIs, these variables can be safely omitted in the UQ to reduce the computational cost. These variables are identified with the Pearson´s $\rho$ and the Sobol index analysis. The selected impactful uncertain variables are forward propagated through the computational model down to the output to obtain the QoIs distributions. Once the output distributions are obtained for the experiments and for the simulations, the differences are quantified using a validation metric. To evaluate these differences, we use the Minkowski $L_1$ norm (MN) validation metric proposed in [9] in two different ways. For the first approach (called $\text{MN}^u$) a uniform distribution is assumed in the experimental data, and the MN integrated between that artificially built experimental empirical cumulative distribution function (ECDF) and the simulation ECDF. For the second approach, so called p-box approach [52], no distribution is assumed in the experimental data. Therefore, the MN is calculated between the simulation ECDF and the maximum ($\text{MN}^+$) and minimum ($\text{MN}^-$) limits of the experimental data.

## Results

Results are split in three parts. Section *Verification* briefly relates the verification results. Section *Sensitivity analysis* shows the results for the SA of the numerical model, which rank the model input variables according to their impact on the outputs. As mentioned in the section *Design and goals of the VVUQ plan*, LHS is used during the SA to sample the input values domain. Results are analysed through scatter plots, Person's ρ correlation and total Sobol indices. Section *Validation with uncertainty quantification* shows a UQ analysis for six validation points reproduced in the SDSU-CS. The six validation points include two different conditions, so called 22[%]@68.42[$bpm$] and 17[%]@61.18[$bpm$], with three pump speeds ($0k$, $8k$ and $11k$ [$rpm$]) each. As the benchtop experiment uses a continuous flow LVAD, we assume there is no beat-to-beat variation. Even if there may be a change in the internal LV flow structures due to its chaotic nature, this does not affect the mass flow through the boundaries or the LV volume.

The model is intended to reproduce inbound and outbound flows in the LV of the CS, therefore the final goal is to correctly reproduce the flow meter signals of the experiment. As the statistical tools for the UQ analysis require scalars, the QoIs chosen to characterise the flows are maximum and average Aortic and LVAD flows. An analysis and comparison of each set of simulation results at every spatial point of the volumetric domain is virtually impossible even for a small number of cases, let alone more than 1000 simulation executions as done in this work. Therefore, even if sample qualitative results are shown for each validation point, the maximum and average flows through the boundaries are calculated and used to calculate statistical trends.

## Verification

This section will briefly describe the results related to the verification credibility factors. Further description of the numerical code verification and numerical calculation verification can be found in the S3 Data.

**Numerical code verification.**   The simulation software used on this manuscript is developed with a continuous integration and continuous deployment (CD/CI) strategy based on git, combining feature-driven development and feature branches with issue tracking. The SQA pipelines ensure continuous integration, running a series of software checks, builds, and regression tests when the developers modify the source code. The tests include a combination of 27 architectures and compilers, optimization options running more than 200 regression tests making a total of more than 4000 different executions. This is complemented with compilation time unity testing and a bi-weekly executed benchmark suite to measure performance evolution.

Numerical code verification is executed as per Section 2 of [12], for a 2D Poiseuille and a 3D Womersley flow problem in a cylindrical tube. These problems have non-trivial analytical solutions that are used as true value. For both cases the discretisation error is monitored as the grid is systematically refined. If the ratio between mesh subdivisions is defined as $r_{i,j} = r_i/r_j$ then, for the cases in this manuscript $r_{1,2} = r_{2,3} = r = 2.0$, a figure considerably larger than 1.3, the minimum value recommended [12]. The largest velocity magnitude root mean square error (RMSE) is 0.03% for the finest mesh and the observed order of convergence $p_{\mathrm{obs}} = 1.907$, compatible with the theoretical order of convergence of 2 of the 2nd order backward differentiation formula (BDF) time scheme used. The largest RMSE is 0.05% for the finest mesh, and the observed order of convergence is $p_{\mathrm{obs}} = 1.82$, close to the theoretical order of convergence of the 2nd order BDF time scheme used. Both, the 2D Poiseuille flow and the 3D Womersley flow are thoroughly described in the S3 Data.

With the result of the code verification process we demonstrate [53] that: *(a)* equations are solved correctly with at most 0.5% error with respect to the analytical solution for the Reynolds numbers representative of LVAD problem; *(b)* observed order of accuracy is similar to the theoretical order; *(c)* The equation coding, transformations and solution procedures are correct.

**Numerical calculation verification.** The original model mesh described in section *Description of the numerical model* was refined twice using the technique described in [54]. The meshes were used for two different configurations of the problem. First, we use a set of simplified boundary conditions. Geometry deformation, valve modelling, and the pump boundary condition are deactivated as the behaviour of these features depend on the CFD results. The goal is to evaluate the CFD solver in a complex domain with a non-ideal mesh. As such geometry does not have an analytical solution, the RMSE are calculated against the finest mesh computed. The maximum velocity magnitude RMSE is 0.91% and the observed order of convergence is $p_{obs} = 0.971$ compatible with the theoretical order of convergence of 1 provided by the first order trapezoidal time integration. The second calculation verification test is executed with the complete model that includes all features. As it is a dynamic problem, time averaged quantities were calculated. The maximum velocity magnitude RMSE is 2.4% and the time-averaged observed order of convergence is $\overline{p_{obs}} = 0.85$ compatible with the theoretical order of convergence of 1 provided by the first order trapezoidal time integration. Detailed results are shown in the S3 Data.

**Discussion of the verification results.** While the applicability of validation results is currently a topic of active discussion [55], the applicability of verification results is rarely discussed. The reason for this is probably the scarce number of verification tests that have a non-trivial analytical solution or a manufactured solution. The tests in this section are executed with Reynolds number close to the ones in the LV, supporting credibility of the solution procedure for an operating condition similar to the validation operating condition.

Simple, stationary physical problems as the 2D Poiseuille flow allow having small errors for a reduced computational cost. When trying to find a solution to more complex transient problems (e.g. the 3D Womersley transient flow) the errors increase. The model solved in this project (section *Description of the numerical model*) is not only a transient problem solved on a fine mesh, but also: *(a)* the fluid mesh is deforming, *(b)* the boundary conditions (such as the valves or the pump boundary condition) vary with the solution of the CFD solver and, *(c)* it requires solving the near-ill-conditioned problem of the valve closing. Therefore, considerably larger errors were expected compared to simpler academic problems. Despite that, the numerical error remained under 2.4% for the most complex use case presented.

## Sensitivity analysis

**Results for the SA.** The SA is intended to identify the input variables with the highest impact in the QoIs. The variable ranges used for the SA are shown in Table 4. The density $\rho$ and the dynamic viscosity $\mu$ are easily and accurately measured. Furthermore, due to the system operation pressures and the fluid bulk properties, these quantities are not expected to change. With this, these two variables are classified as deterministic, knowing their exact value. The rest of the input variables are ranged in approximately one order of magnitude, so the UQ sweeping ranges fall within the SA ranges. To proceed with the LHS a uniform distribution is considered, obtaining 500 samples from the input variables. These samples are used to run 500 simulations, obtaining an ensemble of the QoIs. The sampling and results are shown in the scatter plot at Fig 3 together with the Person's correlation coefficient $\rho$. From a visual analysis of the scatter plot it can be seen that the data is nonlinear, heteroskedastically distributed, and contains multivariate outliers, failing 3 of the 7 assumptions required for Pearson's analysis.

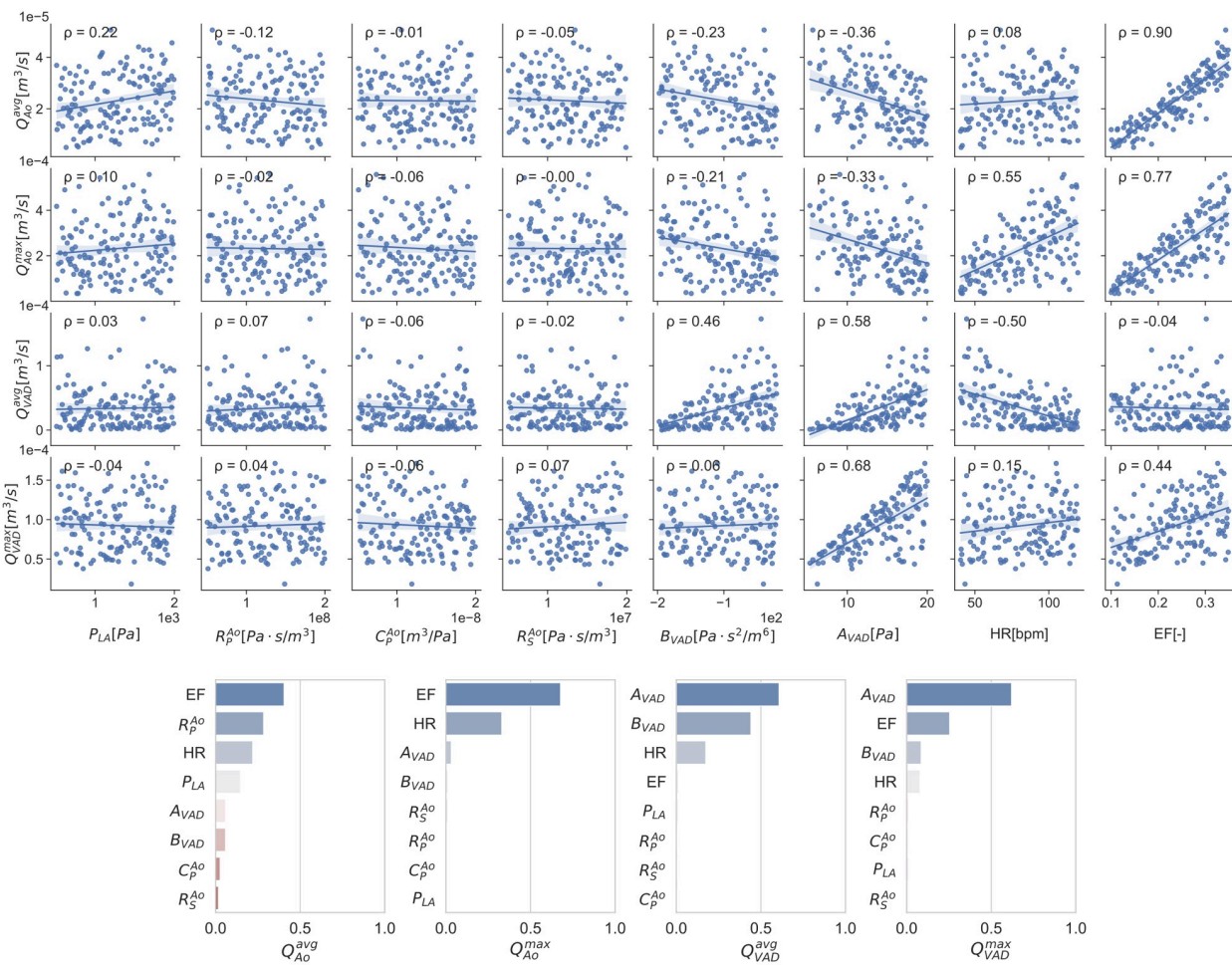

**Fig 3. Scatter plots and total Sobol indices tornado plots for the 8 input variables and the 6 QoIs.** The scatter plot also shows the Pearson's linear correlation number ρ in the top left corner. Units are intentionally avoided in the Y-axis of the total Sobol indices tornado plot to ease its legibility.

To overcome this issue, a global SA is done by calculating total Sobol indices (indicated in section *Sensitivity analysis*). Total Sobol indices provide information of the importance of each input taking into account complex factors like nonlinearities, input interactions, and sample dispersion. The total Sobol index of each input with respect to each QoI are shown as a tornado plot in Fig 3. The larger the index, the more important that input is for the QoI. The total cost of the 500 simulations is about 50 [core-years] in Marenostrum IV supercomputer.

**Discussion of the SA results.** The scatter plots and the Pearson's ρ analysis shown in Fig 3 provide a simple tool to identify the most important variables in the LV-LVAD system. While these tools are useful for a first order approach to understanding the system's behaviour, they fall apart for non-linear effects and complex interactions. Total Sobol indices provide a more insightful tool that accounts for the effect of each input variable and their interactions in each QoI. From the total Sobol indices analysis we can see that the most highly ranked inputs are the EF, HR, $A_{VAD}$ and $B_{VAD}$. Inputs such as the Windkessel parameters $R_P^{Ao}$, $C_P^{Ao}$, $R_S^{Ao}$, and the left atrial pressure $P_{LA}$ have total Sobol indices smaller than 0.25 for at least one QoI so they are qualified as not relevant for the UQ. The reason for this is addressed in the discussion

at the end of this section. The wide ranges chosen for the global SA provide a trustworthy set of Sobol indexes that are applicable to the smaller ranges during the UQ analysis.

While SA is a common tool other fields of cardiac modelling like electrophysiology and solid mechanics [56–58], there is no published work with a local nor a global SA for 3D CFD regarding the LV-LVAD system. Despite this, The trends in Fig 3 agree with experiment data and clinical observations. A higher pump speed, translated as a larger $A_{VAD}$ coefficient, has a positive correlation with the LVAD flow and a negative correlation with the aortic flow [59]. The reason for this is that the suction produced by the pump reduces the aortic valve opening [8, 60]. Also, the HR and EF has a direct positive correlation with the LVAD and aortic flows [61]. The unexpectedly [61, 62] small influence of the mean atrial pressure ($P_{LA}$) and arterial impedance (characterised via $R_P^{Ao}$, $R_S^{Ao}$, and $C_P^{Ao}$) can be explained due to the lack of Frank-Starling mechanism [63] in the silicone ventricle of the experiment and therefore also in its computational analogue. This may raise a concern on the model applicability if it was used for clinical guidance. But, as stated in the CoU of this manuscript: "(. . .) the model is intended to provide a computational replica of a benchtop experiment (. . .)". While the quantification of the so called applicability error would be critical to correctly estimate the total model error, it is still under investigation [64].

## Validation with uncertainty quantification

The global SA in sction *Sensitivity analysis* identified the four most relevant variables, namely the EF, the HR, $A_{VAD}$, and $B_{VAD}$. These are the simulation input variables studied during the UQ analysis. The UQ analysis consist of six validation experiments varying the four chosen inputs. For each validation point, a qualitative set of images is shown that allows visualising the CFD behaviour of the problem. The quantitative results are analysed through scatter plots and ECDFs. To evaluate the differences between the experimental and simulation distributions, we use the MN validation metric already described in section *Uncertainty quantification*. In every figure, experimental results are represented with orange and simulation results with blue.

**Validation points and ranges.** For the UQ analysis, the SDSU-CS is configured at two beating conditions. The condition 22[%]@68.42[*bpm*] has an EF = 22[%] and HR = 68.42 [*bpm*]. The condition 17[%]@61.18[*bpm*] has an EF = 17[%] and HR = 61.18[*bpm*]. Three pump combinations are used in each case, 0*k*[*rpm*] (or pump off with clamped outflow LVAD conduit, recall section *Description of the benchtop model*), 8*k*[*rpm*], and 11*k*[*rpm*]. This makes a total of six validation points as described in Table 5. These six validation points are chosen to vary the QoIs that provide information to answer the question of interest: EF, HR, and pump speed (via the coefficients $a_{VAD}$ and $b_{VAD}$). As there is no information on the precision of the prescribed EF and HR, a 10% error is assumed for these two inputs, producing the ranges in the second and third column of Table 5. Similarly, and as explained in section *Design and goals of the VVUQ plan*, the experiment data accounts for the instrument error. But, as we count

**Table 5. The six validation points used for the UQ analysis.**

| Condition | EF value range | HR value range | Pump speed |
|---|---|---|---|
| 22[%]@68.42[*bpm*] | [19.8,24.2][%] | [65.55, 72.45][bpm] | 0k[rpm] |
| | | | 8k[rpm] |
| | | | 11k[rpm] |
| 17[%]@61.18[*bpm*] | [15.3,18.7][%] | [53,63][bpm] | 0k[rpm] |
| | | | 8k[rpm] |
| | | | 11k[rpm] |

**Table 6. Range of H-Q curve coefficients used for the UQ analysis.** The range is obtained as $2\epsilon_{num} + \epsilon_{fit}$ where $\epsilon_{fit}$ is 10% of the measured value.

| Pump speed | coefficient | Uncertainty classification | characterisation | value range |
|---|---|---|---|---|
| 0k | $a_{VAD}$ | deterministic | exact value | $0.0\ [Pa]$ |
|  | $b_{VAD}$ | deterministic | exact value | $0.0\ [Pa \cdot s/m^3]$ |
|  | $c_{VAD}$ | deterministic | exact value | $0.0\ [Pa \cdot s^2/m^6]$ |
| 8k | $a_{VAD}$ | aleatory | uniform | $[10.91, 12.66] \times 10^3 [Pa]$ |
|  | $b_{VAD}$ | aleatory | uniform | $-[7.90, 7.53] \times 10^7 [Pa \cdot s/m^3]$ |
|  | $c_{VAD}$ | deterministic | exact value | $0.0\ [Pa \cdot s^2/m^6]$ |
| 11k | $a_{VAD}$ | aleatory | uniform | $[19.64, 23.77] \times 10^4 [Ba]$ |
|  | $b_{VAD}$ | aleatory | uniform | $-[9.80, 8.24] \times 10^7 [Pa \cdot s/m^3]$ |
|  | $c_{VAD}$ | deterministic | exact value | $0.0\ [Pa \cdot s^2/m^6]$ |

with only a single experiment per validation point, an error range of 10% is included in the QoIs to account for the measurement uncertainty. To calculate one of the validation metrics shown ($MN^u$) we assume a uniform distribution in the measured QoIs for that assumed range. On the contrary, the multiple simulations executed let us calculate the ECDF used for the metrics. The method to calculate the variable ranges for the pump model inputs is explained in the S4 Data. The coefficients range and the uncertainty characterisation are shown in Table 6.

These simulation variables are sampled using a LHS obtaining 50 samples per validation experiment, making a total of 300 numerical simulations that required about 30 [core-years] in Marenostrum IV supercomputer.

**Condition 22[%]@68.42[*bpm*].**    Fig 4 shows qualitative surface results for the three pump speeds for the condition 22[%]@68.42[*bpm*] and the pump speeds 0*k*, 8*k*, 11*k*[*rpm*]. Afterwards, Figs 5 to 7 provide quantitative results for the referenced condition and pump speeds. The flow plots in Figs 5a, 6a and 7a show the aortic ($Q_{Ao}$) and LVAD flow ($Q_{VAD}$) for the experiment (orange) and the simulation (blue). As the UQ analysis also accounts for HR, the time axis is normalised. The scatter plots in Figs 5c, 6c and 7c show the inputs in the x-axis and the outputs in the y-axis for the experiment (orange) and the simulation (blue). The orange range in the y-axis is representing the assumed 10% measurement error and the simulation results kernel distribution estimation (KDE) is represented in blue shades surrounding the simulation measures. Figs 5d, 6d and 7d show the simulation ECDF in blue and the experimental limits with two vertical orange ranges, together with the artificial uniform distribution used to calculate $MN^u$. Finally, the bench and numerical experiments data limits and the multiple MN are shown in Figs 5b, 6b and 7b.

**Condition 17[%]@61.18[*bpm*].**    Results are presented similarly to section *Condition* 22 [%]@68.42[*bpm*]. Fig 8 show a set of time frames for the three pump speeds during the condition 17[%]@61.18[*bpm*]. Afterwards, Figs 9 to 11 show the experimental and simulation results. Again, the experimental results are shown in orange and simulation results in blue. the experiment and simulation ranges and the validation metrics are shown in Figs 9b, 10b and 11b.

**Discussion of the UQ results.**    Similarly to the SA, while UQ analyses have been recently done for electrophysiology and electromechanical models of the heart [56, 58, 65], but no equivalent studies have been conducted for ventricular CFD.

The first noticeable feature in the scatter plots is the lack of dispersion in the x-axes for the experimental results. The x-axes represent the prescribed values of the inputs in the experiment or simulation. As there is a single execution of the experiment per validation point, there is no information on the input variable distribution. This fact that translates as zero dispersion

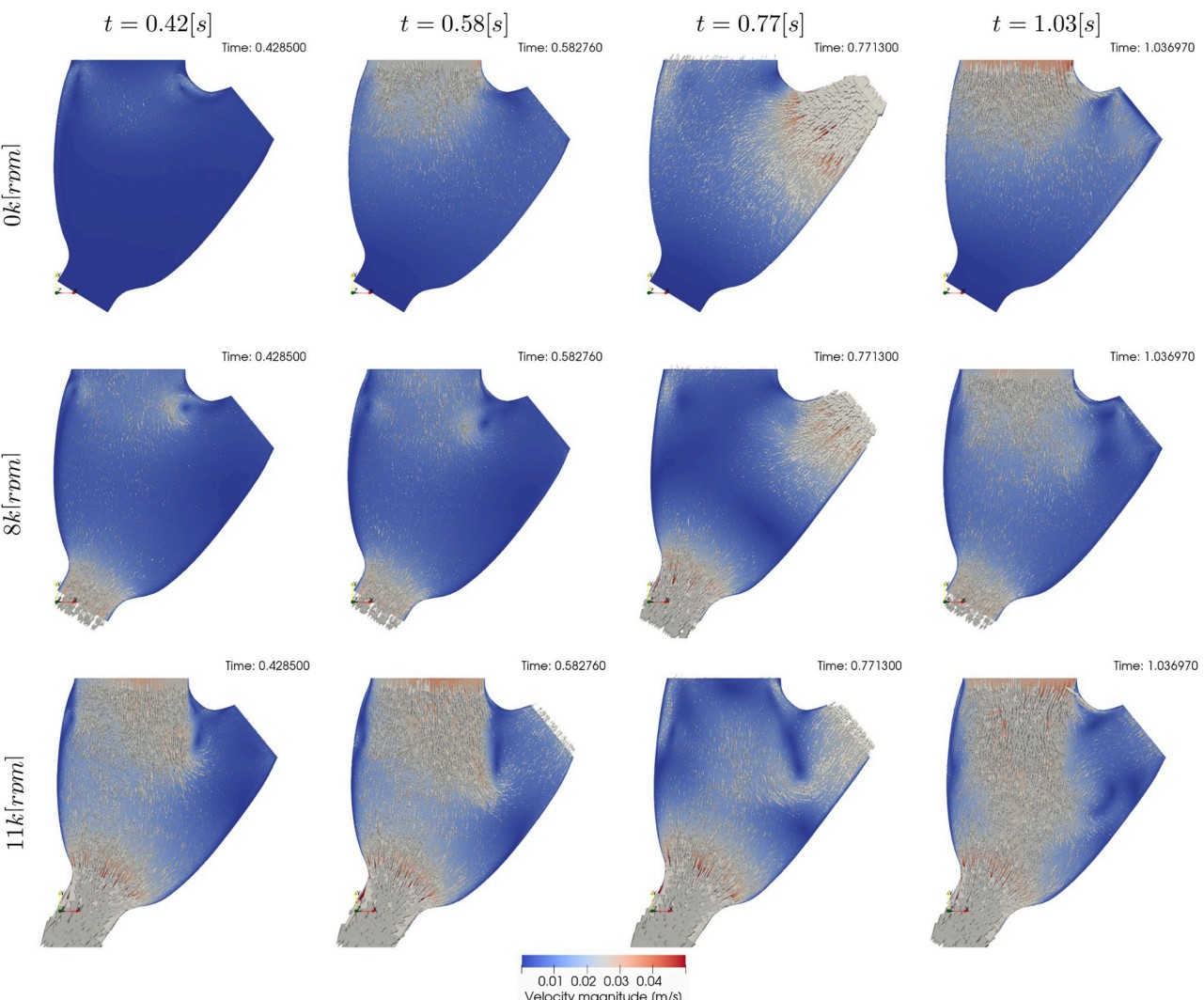

**Fig 4. Qualitative surface results for the condition 22[%]@68.42[bpm].** Pump speed is $0k[rpm]$, $8k[rpm]$, and $11k[rpm]$ in the first, second, and third rows respectively. The columns indicate different time frames in the simulation. $t = 0.42[s]$ show results for the plateau previous to systole, $t = 0.58[s]$ show the atrial kick, $t = 0.77[s]$ show systole, and $t = 1.03[s]$ shows diastole. Videos of the simulations can be found in the S1 Video.

in the x-axis at experimental scatter plots. On the contrary, the multiple beats contained in each validation point and the measurement error in the flow meter is seen as y-axis dispersion for the experiment. Even with a highly reproducible and tested experimental setup as the SDSU-CS we were not able to obtain the experimental probability distributions required for the highest rankings in the ASME V&V40 standard. To be able to obtain these distributions, the exact same experiment should be repeated multiple times, a time consuming and expensive process. While working with these distributions as input data is unarguably optimal, we face the most common case where only a single experimental data point is available [9].

Another noteworthy detail is the fact that the MN is an absolute metric, therefore its interpretation depends on the QoI's range and mean value at the specific condition. As an example, the $0k[rpm]$ cases for both conditions may seem the trivial solution for $Q_{VAD}^{avg}$ and $Q_{VAD}^{max}$. but, even if they have the smallest validation metrics in this manuscript, there is no overlap for these QoIs in the scatter plots.

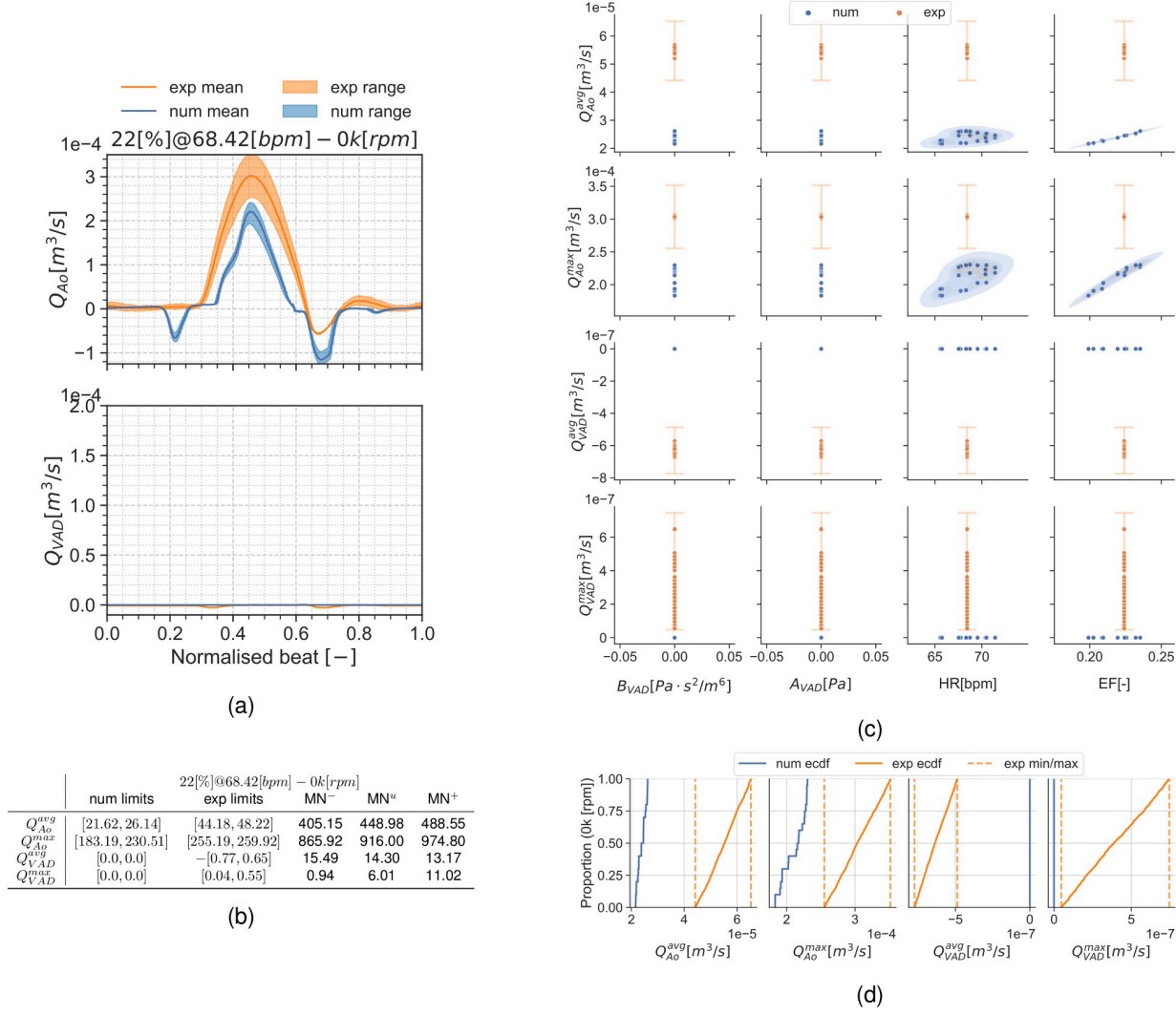

**Fig 5. Summary for the condition 22[%]@68.42[bpm] and 0k[rpm].** (a): aortic valve and LVAD flows. (b): validation metrics. (c): scatter plot showing the experimental and simulation data. (d): ECDF for the simulation, experimental data limits and the constructed uniform distribution.

Results show the smallest validation metrics (i.e. better agreement) for the mid-point working conditions with larger differences for the extreme cases. The large uncertainty ranges in the pump H-Q curve (see S4 Data and Table 6) for the 8$k[rpm]$ and most noticeably 11$k[rpm]$ speeds produce a considerable dispersion in the simulation results. This dispersion is most noticeable in the 11$k[rpm]$ cases (Figs 7 and 11) as a large uncertainty range in the flow curves (Figs 7a and 11a), but it also exhibits itself in the empirical cumulative distribution function (ECDFs) (Figs 7d and 11d) and the scatter plots (Figs 7c and 11c) as a poor overlap of the blue-shaded simulation KDE and the orange range for the experiments. Particularly, Fig 11c shows a clustering in the $Q_{Ao}^{avg}$ and $Q_{Ao}^{max}$ simulation points. As part of the application study in this manuscript, we show in Fig 12 the average aortic valve flow $Q_{Ao}$ for each execution. The data has a threshold at $5 \times 10^{-6}[m^3/s]$ (0.3[L/min]), moment where the aortic valve starts to fully open allowing a consistent flow through it. This is because the $a_{VAD}$ and $b_{VAD}$ uncertainty ranges are so wide that they even allow aortic valve opening for some scenarios, something

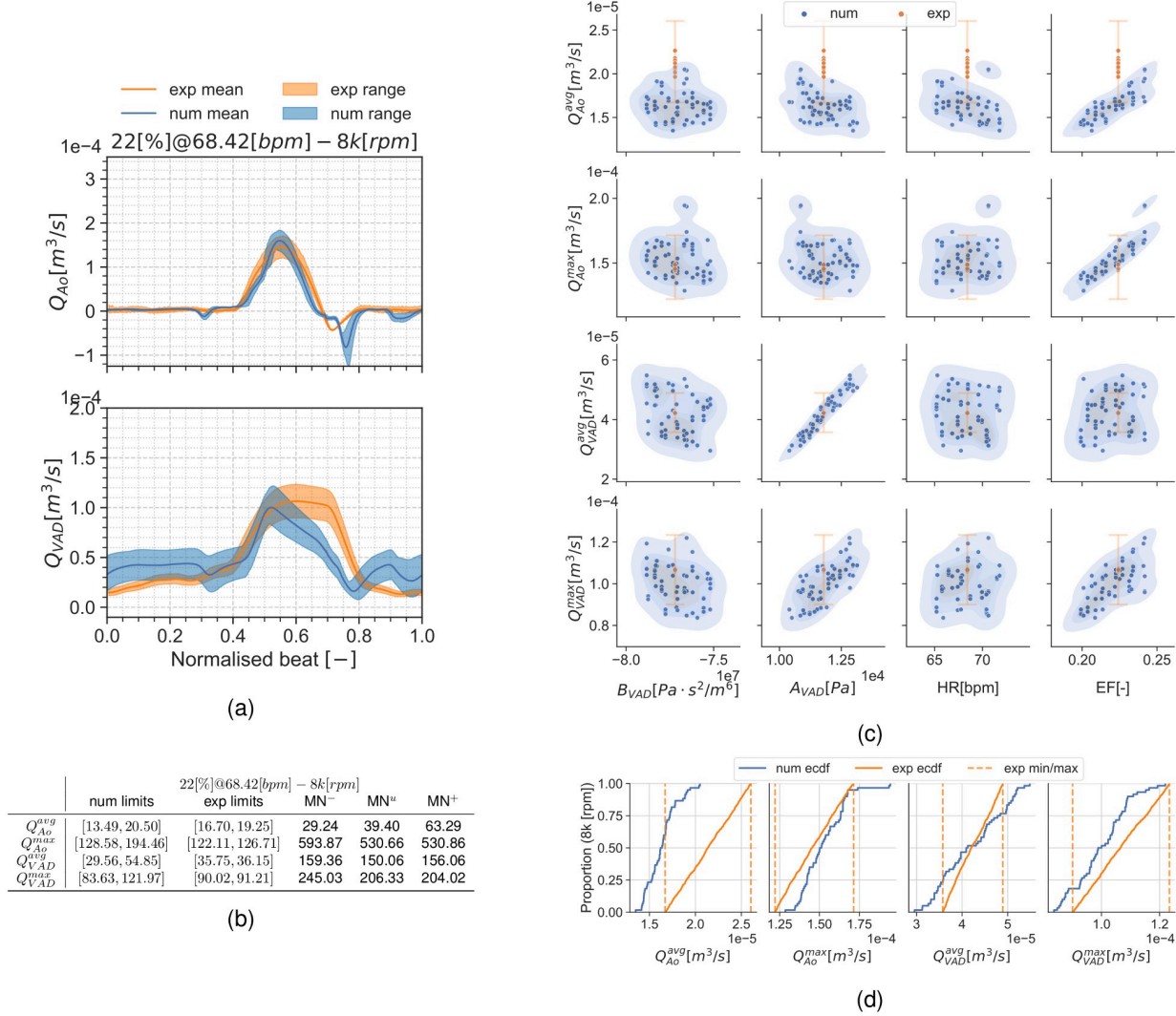

**Fig 6. Summary for the condition 22[%]@68.42[bpm] and 8k[rpm].** (a): aortic valve and LVAD flows. (b): validation metrics. (c): scatter plot showing the simulation and experimental data. (d): ECDF for the simulation, experimental data limits and the constructed uniform distribution.

also observed in [59]. The largest validation metric (Figs 7b and 11b) with the pump operating at $11k[rpm]$ is for $Q_{Ao}^{max}$ for both conditions $17[\%]@61.18[bpm]$ and $22[\%]@68.42[bpm]$.

The $0k[rpm]$ results at Figs 5 to 9 also show a mismatch between the numerical and the bench data. As the pump H-Q curves are forced to zero in the simulation, there is also zero uncertainty in the associated inputs ($a_{VAD}$ and $b_{VAD}$). This produces unequivocally LVAD flows equal to zero $Q_{VAD}^{avg} = Q_{VAD}^{max} = 0.0[m^3/s]$ and a Dirac's delta probability distribution function for these QoIs in the simulation results. On the contrary, the experiment still shows a small y-axis scattering in the LVAD QoIs (Figs 5c and 9c). This is produced due to flow disturbances around the flow meter and the sensor's offset error (see flow-meter characteristics in section *Description of the benchtop model*).

The $0k[rpm]$ cases also highlights a modelling error in the simulation results, most clearly in the flow curves in Figs 5a and 9a: the aortic flow wave produced by the pressure curve in the model is too triangular and short-timed, and the backflow during the valve closing is too large. These differences may have been introduced by the simplified aortic valve model. Despite the

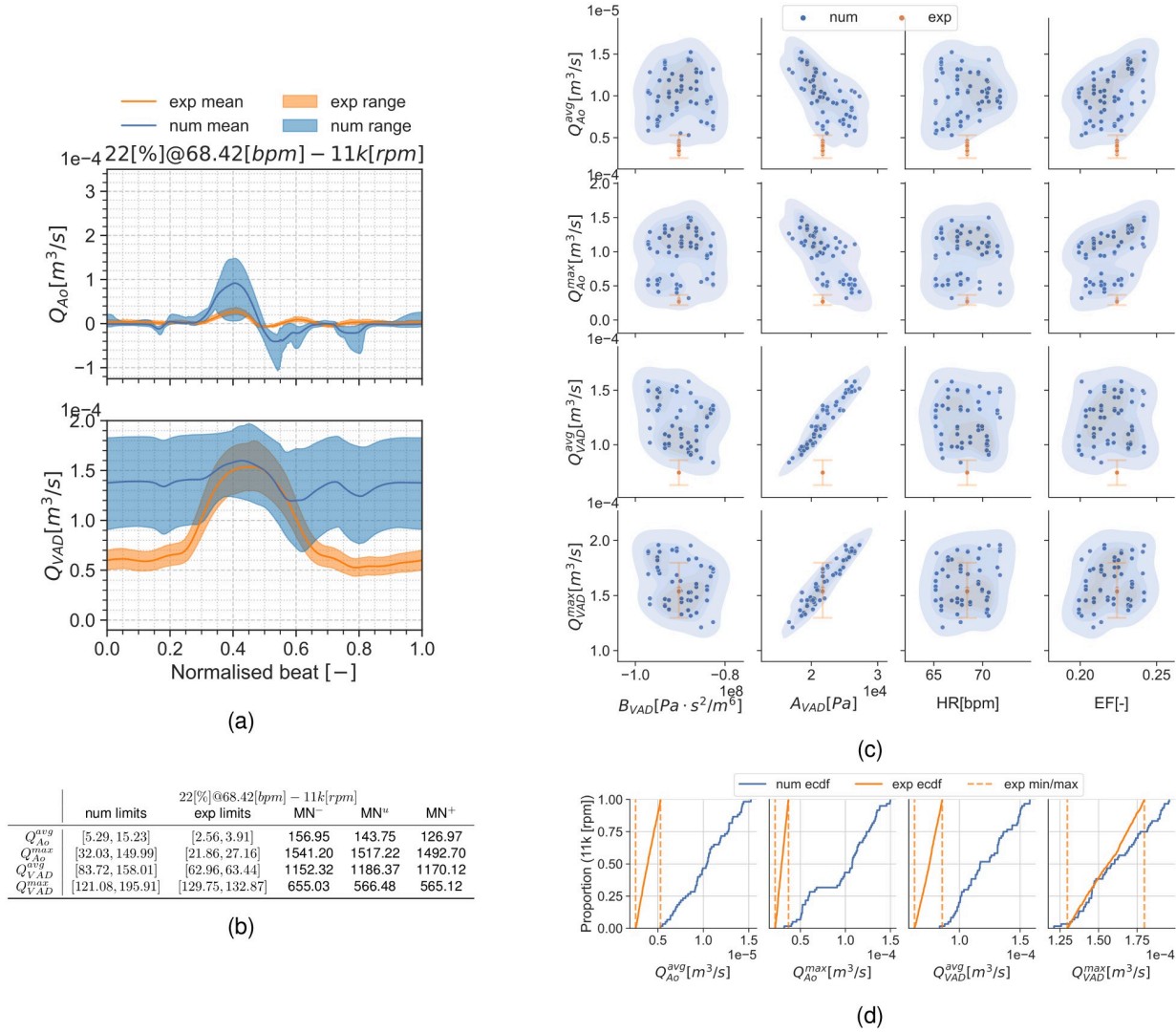

**Fig 7. Summary for the condition 22[%]@68.42[bpm] and 11k[rpm].** (a): aortic valve and LVAD flows. (b): validation metrics. (c): scatter plot showing the simulation and experimental data. (d): ECDF for the simulation, experimental data limits and the constructed uniform distribution.

poor overlap for the distributions seen in the ECDF plots in Figs 5d and 9d and the scatter plots in Figs 5d and 9d, the validation metrics in Fig 5b and 9b for $Q_{VAD}^{avg}$ and $Q_{VAD}^{max}$ are almost negligible as the values enforced in the simulation are close to the experimental ones.

The $8k[rpm]$ operation condition provides the smallest overall metrics (i.e. best agreement) between the experimental and simulation. For the condition $22[\%]@68.42[bpm]$ this can be seen as a good overlap between the experimental ranges and the simulation distributions. Similarly, the simulation ECDF curves (Fig 6d) overlap the experiment and numerical ranges for every QoI. Almost similarly, the condition $17[\%]@61.18[bpm]$ shows an overlap for most variables in the ECDF curves (Fig 10d) except for those associated with the aortic flow ($Q_{Ao}^{avg}$ and $Q_{Ao}^{max}$). Despite that lack of overlap, the validation metrics in Fig 10b are also reduced.

The lack of publications with UQ analyses for ventricular CFD makes comparing these results a difficult task. The results suggest a mismatch between the experimental and

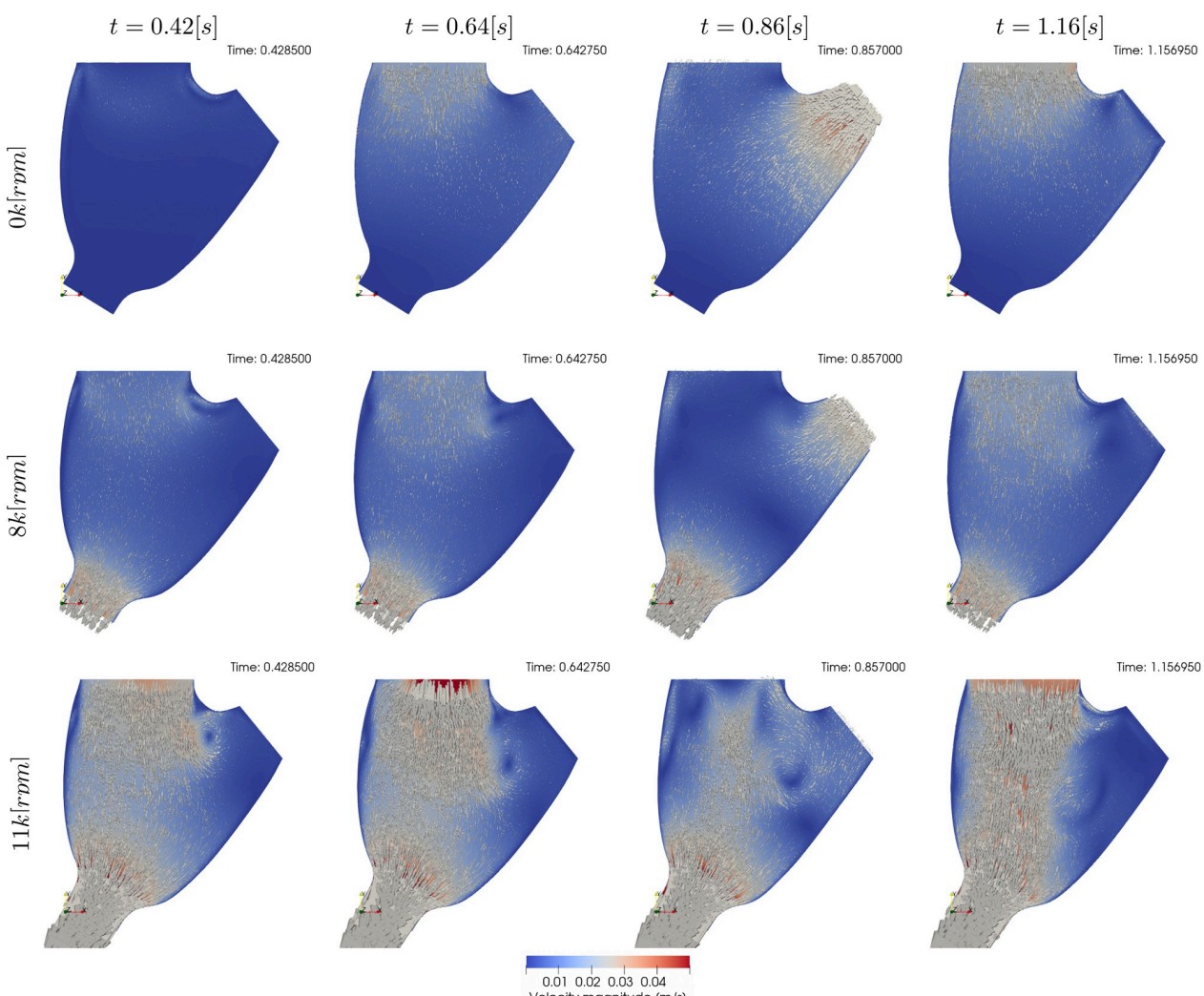

**Fig 8. Qualitative surface results for the condition 17[%]@61.18[bpm].** Pump speed is $0k[rpm]$, $8k[rpm]$, and $11k[rpm]$ in the first, second, and third rows respectively. The columns indicate different time frames in the simulation. $t = 0.42[s]$ show results for the plateau previous to systole, $t = 0.64[s]$ show the atrial kick, $t = 0.85[s]$ show systole, and $t = 1.16[s]$ shows the diastolic filling. Videos of the simulations can be found in the S1 Video.

simulation results for the $0k[rpm]$ (LVAD off) and the $11k[rpm]$ cases. These validation points highlight potential issues that are worth exploring.

## Discussion on the V&V40 credibility factors: Achieved score

This section is intended to summarise the achieved scores for the credibility factors described in the ASME V&V40 [11] and summarised in Table 3. The section is intended to compare the pre-selected goal with the final achieved score. For the reader's reference, the translation from the scores to the required activities can be found in the S1 Data. The rationale behind the chosen goal for this work can be found in the S2 Data. In the rest of this section we will summarise the rationale behind the scores obtained for each credibility factor.

**Section 1: Verification credibility factors.** For the calculation and code verification the ASME standards define a set of rankings that ranges from no verification up to an extensive

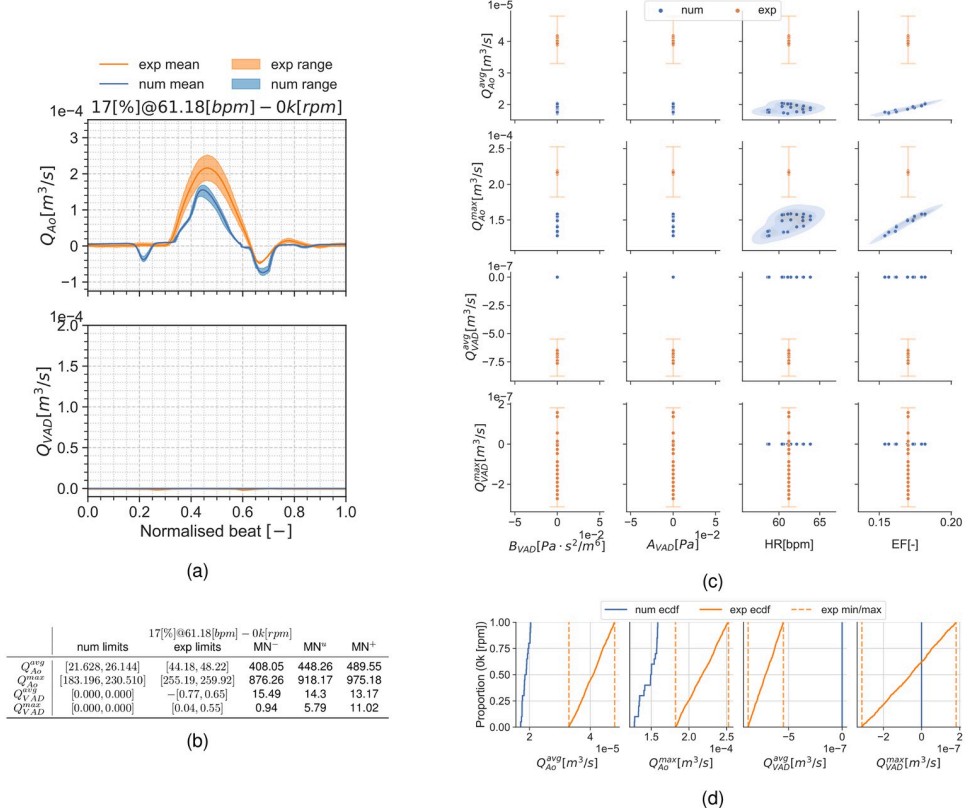

**Fig 9. Summary for the condition 17[%]@61.18[$bpm$] and 0$k$[$rpm$].** (a): aortic valve and LVAD flows. (b): validation metrics. (c): scatter plot showing the simulation and experimental data. (d): ECDF for the simulation, experimental data limits and the constructed uniform distributions.

set of tests. The code used as simulation engine follows rigorous SQA enforced by the code life cycle tool. Also, the fact that the simulation code is partially open source and part of the Partnership for Advanced Computing in Europe (PRACE) Unified European Applications Benchmark Suite (UEABS) ensures the code transparency and constant scrutiny [54, 66, 67]. Added to this, two numerical code verification tests and three numerical calculation verification tests (S3 Data) were executed to ensure code correctness and bounded numerical error for this problem. Comparing the executed tasks with the rankings in the S1 Data, led us to rank the SQA, NCV, discretisation error, and numerical solver error with the maximum score, surpassing the original desired goal. Due to the lack of external manpower, the inputs of the solver where only checked by internal review. This led us to achieve a C out of a maximum D in the user error credibility factor. As the number of input variables is rather small, the original goal was B out of D and therefore the original goal is achieved.

**Section 2: Validation credibility factors.** While the background model is based on the well known Navier-Stokes equations, there are multiple associated sub-models like the pump H-Q performance curve model, the lumped valve model or the Aortic impedance Windkessel model. Some of the assumptions for the simplification on the UQ were tested *a-priori* during the SA, therefore the model form correctness scores a B out of C, the desired goal. On the model input credibility factor, due to the thorough SA and UQ executed we achieved the desired goal of B out of C. The analysis was executed using retrospective experimental data, which was not gathered for VVUQ use. A single silicone ventricle was used as test sample,

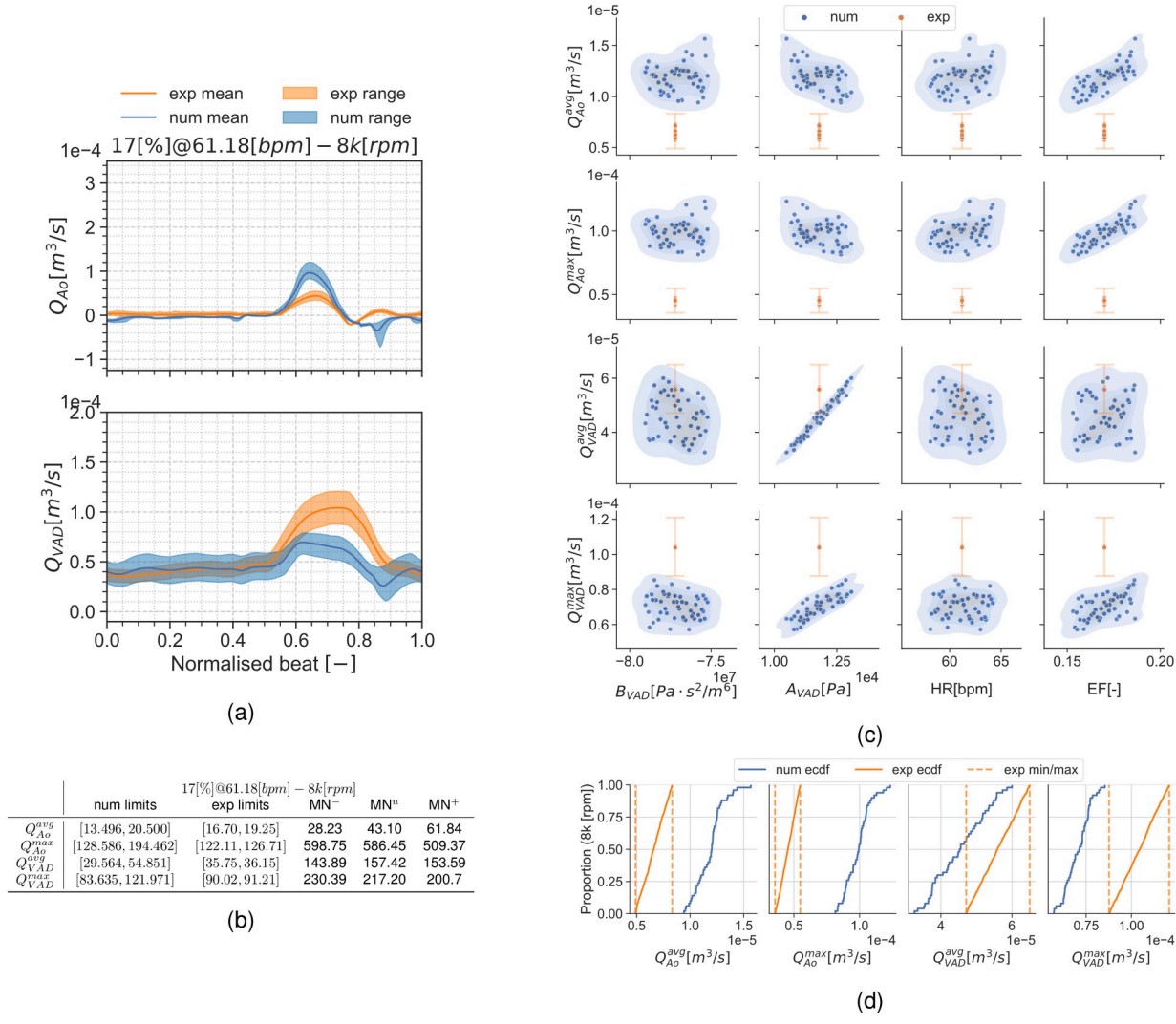

**Fig 10. Summary for the condition $17[\%]@61.18[bpm]$ and $8k[rpm]$.** (a): aortic valve and LVAD flows. (b): validation metrics. (c): scatter plot showing the simulation and experimental data. (d): ECDF for the simulation, experimental data limits and the constructed uniform distribution.

achieving an A out of C for the quantities of test samples credibility factor and an A out of D for the range of the test sample characteristics credibility factor. Despite this, the geometry used was characterised in all its features with a computer drawing tool, obtaining a score of C out of C. The CoU focuses on reproducing the bench experiment and assessing mass flow through the boundaries and not on analysing the internal LV fluid dynamics. Therefore it does not require evaluating the QoIs for multiple geometries. With this, a single idealised geometry was considered for the credibility evidence. With this, all the goals for the test sample credibility factors were achieved or surpassed. Further rationale can be found in the detailed table in the S2 Data. On the test conditions credibility factor, multiple test conditions were tested (achieving a B out of B) representing the expected extreme conditions range (achieving a C out of D), measuring all the key test conditions (achieving a B out of C). Despite this, the uncertainty of the test conditions was not characterised, achieving an A out of C. Most of the goals for the comparator credibility factor where achieved or surpassed, except for the characterisation of the test condition uncertainties. Finally in the assessment credibility factor, the

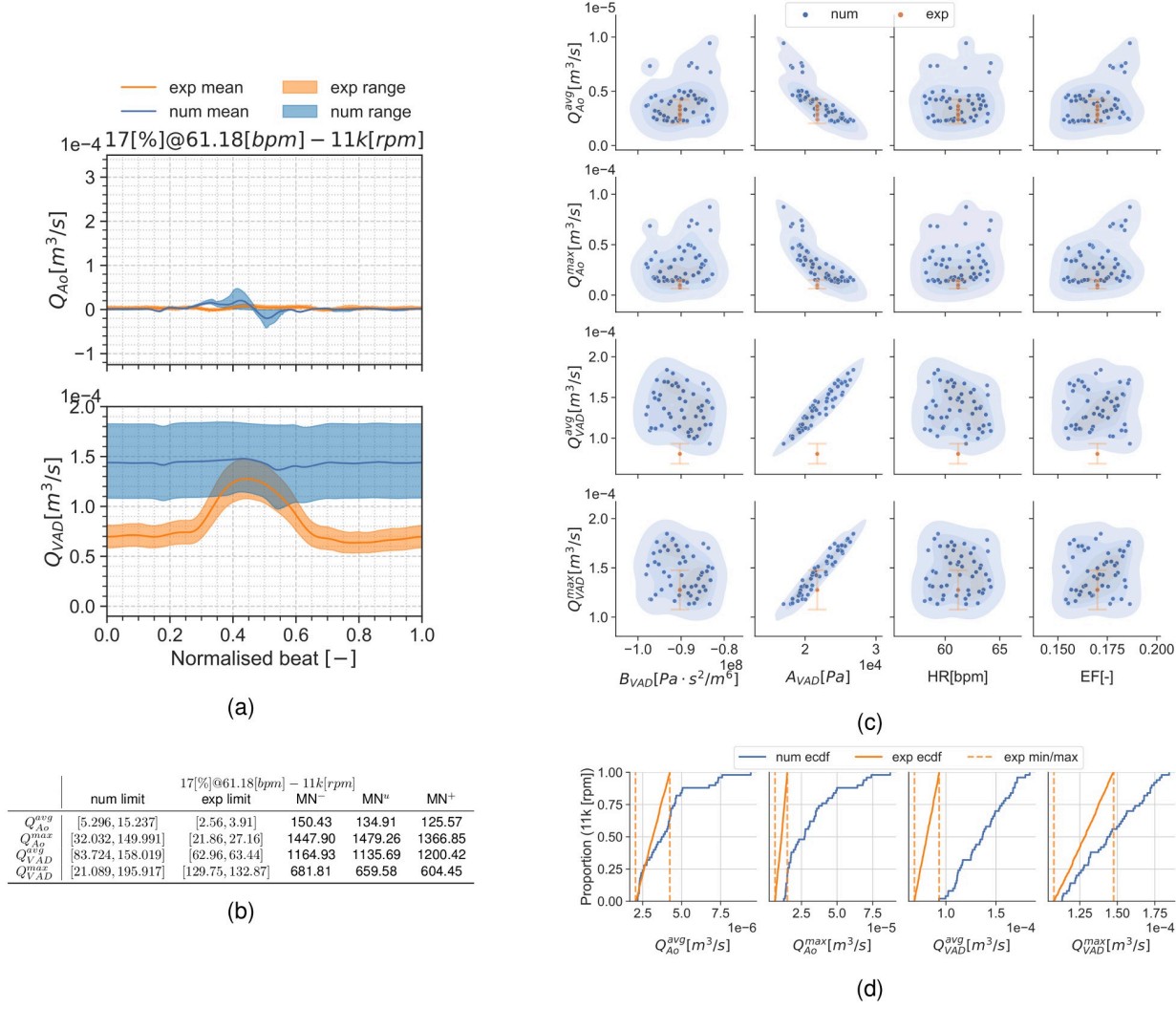

**Fig 11. Summary for the condition $17[\%]@61.18[bpm]$ and $11k[rpm]$.** (a): aortic valve and LVAD flows. (b): validation metrics. (c): scatter plot showing the simulation aned experimental data. (d): ECDF for the simulation, experimental data limits and the constructed uniform distributions.

accuracy of the simulation output is evaluated. As the simulation is designed to reproduce the experiment physics and resulting QoIs, all the input types and ranges of all inputs were identical,achieving a C out of C for the equivalence of inputs credibility factor. Also multiple outputs were rigorously compared via two approaches of validation metrics, achieving a C out of C for the equivalence, rigour and quantity of output variables credibility factor, surpassing the predefined goals in all of them. As the level of agreement was satisfactory for some key comparisons, we achieve the desired goal of B out of C for the agreement of output comparison credibility factor.

**Section 3: Applicability credibility factors.** This item assesses how relevant the validation results are to the CoU. As the CoU proposes the tool for analysing LVAD and Aortic valve flow to analyse total cardiac output and aortic valve opening, we assign the maximum rank C out of C for the relevance of the QoIs for the question of interest. As the number of validation points could be increased to target a larger operation envelope, the validation activities only

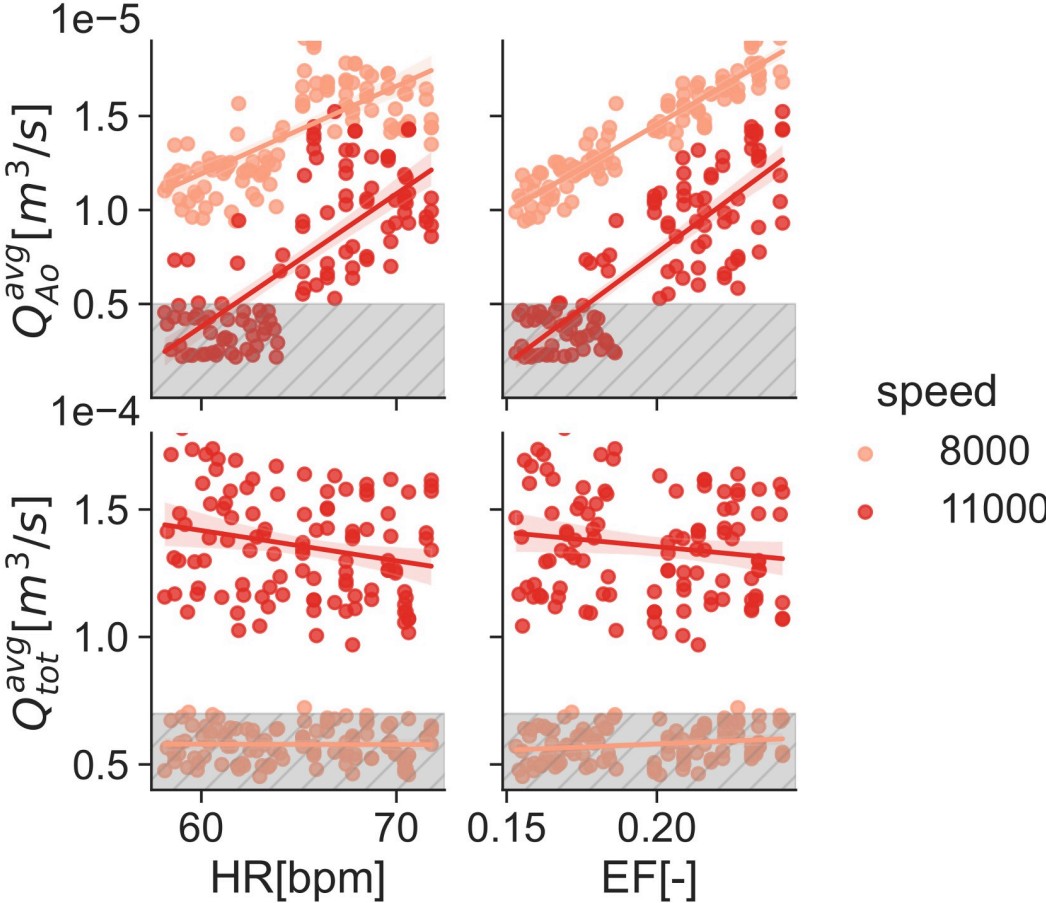

**Fig 12. Aortic valve flow $Q_{Ao}^{avg}$ and total flow $Q_{tot}^{avg} = Q_{Ao}^{avg} + Q_{VAD}^{avg}$ as a function of the HR and EF.** Data is shown for both, $8k[rpm]$ and $11k[rpm]$. The grey hatched region represents the minimum limit for both $Q_{Ao}^{avg}$ and $Q_{tot}^{avg}$.

encompassed some validation points for the CoU, achieving a C out of D. In both cases, the goals were surpassed.

**Overall credibility assessment.** We demonstrate the model to be sufficiently close to the validation points for the simple CoU proposed. While some of the credibility factors did not obtain the maximum achievable score, they did obtained or surpassed the desired goal for the medium risk application (refer to Table 3 for a summary). Riskier applications, where the numerical model drive safety related conclusions or where the final decision relies more on middling, would require achieving those maximum achievable scores. Further improvements on the model to obtain these maximum scores are discussed in the conclusion.

## Application to ramp study

While the CoU constrains the usage of the numerical model for preclinical development of LVADs, we show a use case outside of that CoU where the model presented in this manuscript can be useful. The applicability of the credibility evidence for the proposed use case is discussed at the end of this section. As explained earlier, a ramp study [7] is routinely performed after LVAD implantation to select the pump speed for the patient. The selection is based on LV flow and geometry measured with echocardiography while the LVAD speed is increased over a wide range. The optimal speed is chosen to ensure end-organ perfusion. The desired

cardiac output (measured as $Q_{tot}^{avg} = Q_{Ao}^{avg} + Q_{VAD}^{avg}$) is dependent of the patient BSA, and about $3.6\times10^{-6}[m/s]$ ($2.2[L/min/m^2]$) per unit of BSA. This is to say, for an average BSA of $1.9[m^2]$, the desired $Q_{tot}^{avg}$ is $7\times10^{-5}[m^3/s]$ ($4.2[L/min]$). Bowing of the intraventricular septum towards either RV (at low LVAD speeds) or LV (at high speeds) is avoided, particularly if the LVAD inflow cannula is oriented towards the AoV. On the contrary, AoV opening increases at lower LVAD speeds. It is desirable to achieve opening of the AoV (measured via a $Q_{Ao}^{avg} > 5 \times 10^{-6}[m^3/s] = 0.3[L/min]$), at least intermittently, to avoid aortic regurgitation. The balance among these considerations are determined by the clinicians present, and the speed selected for long term LVAD support. Interestingly, most LVAD patients experience changes in cardiac geometry and function during the use of LVAD support. For example, increased heart rate or blood pressure can result in shorter systolic durations and alter AoV opening. Reduction in LV volume due to reverse remodeling and improvements in ejection fraction may shift the intra-ventricular septum position or enable greater AoV opening. With this, the final pump speed will depend on the patient's hemodynamic condition, quantified here via the EF and HR. In all of these examples, having a more adaptive system begins with a validated tool such as the model proposed herein.

The model results extend the range of the experimental studies by evaluating the QoIs over a range of heart rate and ejection fraction, specifically AoV flow and total aortic flow ($Q_{Ao}^{avg} + Q_{VAD}^{avg}$). Fig 12 shows that the model predicts a small but notable decrease in LVAD flow with increasing HR and EF, which is accompanied by an increase in flow through the AoV. Fig 12 shows that the lower pump speed limit is bounded by $Q_{tot}^{avg}$ and the upper pump speed limit is bounded by $Q_{Ao}^{avg}$. At $8k[rpm]$ the pump is unable to meet the $Q_{tot}^{avg} > 7 \times 10^{-5}[m^3/s]$ ($4.2[L/min]$) requirement for the range of HR and EF analysed. Oppositely, the $11k[rpm]$ case is unable to meet the $Q_{Ao}^{avg} > 5[cm^3/s]$ requirement for the situations with HR≲ $65[bpm]$ or EF≲ $18[\%]$. This agrees with the findings in [7] showing that the AoV closes at $9124 \pm 1,222$ $[rpm]$ with an optimal LVAD speed of $8850 \pm 470[rpm]$.

While it is exciting to show a clinical application case as the one described here, the CoU presented in this manuscript specifies that the model may only be used during preclinical design of the device. With the current CoU, the credibility evidence gathered is not complete enough for a clinical application. The simulation tool and the bench experiment contain a set of simplifications that should be studied in a future CoU addressing clinical practice. While the experimental and numerical models use an idealised and smooth LV, patient´s LV geometries come in a vast variety of volumes, shapes and levels of trabeculations. Human left ventricle (LVs) are also mechanically connected to the RV and the surrounding tissues, which affect the LV behaviour. In this work an homogeneous LV contraction is assumed, which may not be valid for advanced HF patients with large non-contractile regions. Future iterations of this work addressing clinical CoUs should evaluate these variables in the bench-top experiment and numerical model to safely use the listed simplifications in the current version of the proposed tool.

Ideally, in future iterations of LVAD therapy, the device will be able to remotely monitor the HR, and the speed setting updated to adjust for the patient condition. With improvements in remote monitoring, device adjustment, the presented model could serve as a baseline tool for a smart interface between the patient's LVAD and their clinical team, and form the foundation for automated speed control.

## Conclusion

This manuscript provides a thorough detail and execution of a VVUQ plan of a clinically relevant numerical model. Starting from a major concern of LVAD treatment we define the

V&V40 terminology and goals following the approved standard, and proceed to design a VVUQ plan that we afterwards execute and analyse with statistical tools. While [68] reviews the use of computer model for critical health applications under the ASME V&V40 standard [11], this work presents for the first time a complete execution of a VVUQ plan following the cited guideline.

Even if simpler LVAD numerical models have been published in the past, this is the first including a deformable ventricle, a pressure driven valve model and a dynamic LVAD boundary condition. The numerical model has been created to faithfully reproduce the SDSU-CS. To do so, the numerical model required to deform the mesh with the same pattern as the experiment, a 0D model of the systemic arteries, a novel approach for the valves that is driven by the transvalvular pressure gradient, and a novel approach to represent the LVAD through an H-Q curve performance function as boundary condition.

Moreover, such a model has been subject to the V&V40 pipeline allowing to bound the uncertainties in the simulation. The main facilitator for this has been the usage of a bench experiment as a source of comparators. When comparing animal experiments with benchtop experiments, the former have a larger inter-subject variability, lower reproducibility and lower access to the QoI, while the latter provide a more reproducible and accessible set of comparators. To ensure the solution procedure correctness, the numerical model was subjected to two code verification tests that bounded the numerical error for an operating condition close to the validation points. The two calculation verification tests executed, provided a measurement of the uncertainty produced by the spatial discretisation. The local SA provided a graphical understanding of the model's behaviour, while the global SA based on total Sobol indices highlighted the most impactful input variables. This variable reduction brings the consequent reduction in the UQ analysis computational cost. The six validation points swiped through three pump speed velocities, two EF and two HR. The final use of the model is the application to the ramp study [7]. The model predicted an operational pump speed range that allows obtaining aortic valve opening and the desired total aortic flow. Moreover, the pump speed ranges predicted by the model agrees by the ranges found in [7]. From an applicability perspective, the impact of modeling and simulation is only achieved when credibility goals and resources align. While the execution of a thorough VVUQ plan provides the final results high credibility, it is a time consuming and computationally expensive process. Achieving the highest scores in Table 3 requires not only rigorous testing of the simulation code but also a large number of experiment executions. Given the ASME V&V 40 risk-based credibility assessment, the burden associated with the highest scores is only required for high risk applications. Unfortunately, these applications are where the simulations will prove the most beneficial. Considering the level of effort required to surpass the credibility threshold defined by the ASME V&V40 standard, it is arguable that the minimum credibility threshold for the computational models has been set up considerably higher compared to the credibility requirements for benchtop experiments. However, a detailed discussion on the costs and relative reliability of different models (clinical, animal, bench, simulation) given current best practice for each, is out of the scope of this manuscript.

Historically, the regulatory entities have accepted bench experiments, animal experiments, and human trials as sources of evidence for pre-market approval (PMA) applications [69]. The advent of numerical models into biomedical devices design raises new questions, especially if their use is intended to evaluate the device safety and effectiveness during the regulatory evaluation. Even if widely accepted, bench experiments provide an insight into the device performance for generally simplified geometries and under strictly controlled conditions. Despite this, they allow evaluating the device in an environment similar to the usage conditions. Similarly, animal experimentation has been a cornerstone of the regulatory system for decades. Despite this, the translation of animal experiments to humans has been widely criticised

[70–73] due to three main reasons: *(1)* the effects of the laboratory environment and other variables on study outcomes, *(2)* disparities between animal models of disease and human diseases, and *(3)* species differences in physiology and genetics [70]. While human clinical trials provide the only true reflection of the device behaviour in the intended usage, they are constrained to tightly regulated ethical standards and small samples. As the research questions become more sophisticated, it is becoming increasingly difficult to find trustworthy answers with the limitations of the available sources of evidence [74]. Using numerical modelling as a source of evidence opens a new door for the regulatory process with the promise of resolving the multiple flaws [70–74] present in the classical approach. With such pledges, it is expected that numerical models are subject to a tight credibility scrutiny. But, answering how much we can trust simulations and in what portion they will be replacing bench and animal experiments, or even human trials is something only history will tell.

Being the first iteration of the VVUQ plan, the results highlighted a number of items to be improved in the future. On the bench experiment side, including multiple executions for each validation point, would improve the validation metrics in the UQ, as the input variables could be characterised with a probability distribution instead of a forced 10% experimental uncertainty as done in the current manuscript. Given the hypotheses in this manuscript, we did not tested the effect of the LV bag size and shape, although that would increase the application range of the model. Also, retrieving multiple measurements of the H-Q curve of the LVAD for the operating condition will also have a direct positive impact on the final validation metrics, as we can conclude from the results in this work. The large uncertainty ranges in the H-Q curves in this manuscript produce distributions in the QoIs. Using a lumped H-Q curve representation for the LVAD behaviour may also have an impact on the results. While a quadratic approximation of the pressure-flow relation in a pump is a common engineering practice, it is still a simplification that may fall apart in complex use conditions. On the simulation side, the pressure-driven porous layer is certainly affecting the LV vortical structures, as the valve geometries shapes the LV flow patterns [23, 31, 32]. Adding valve geometries will improve the intra-LV flow patterns in future applications where vortex quantification becomes critical. Also, including the deterministic variables in the SA will increase the model credibility, necessary for a higher credibility score in the model form factor. These improvements will not only make a more accurate model, but also increase the scoring in the credibility factors. With these improvements, the model could target riskier applications.

## Supporting information

**S1 Video. The video shows an example simulation for each validation point.** The top row shows the $17[\%]@61.18[bpm]$ condition and the bottom row the $22[\%]@68.42[bpm]$ condition. The different columns are the results for $0k$, $8k$, $11k[rpm]$. The surfaces are coloured by velocity magnitude and the arrows show the flow direction.
(MP4)

**S1 Data. Ranking for risk informed credibility assessment.** This section provides the gradiation for each credibility factor and actions required on each item in the standard ASME V&V40.
(PDF)

**S2 Data. Rationale behind the achieved scores for each credibility factor.** This section provides the rationale behind every achieved score for each credibility factor. The score is taken from S1 Data.
(PDF)

**S3 Data. Verification evidence.** Results of the verification tests.
(PDF)

**S4 Data. Calculation of the pump input variable ranges.** Experimental data and coefficient fitting for the pump parameters.
(PDF)

## Acknowledgments

The authors want to extend their gratitude to Simone Venturi from Sandia National Laboratories and Tim Baldwin from the U.S. Food and Drug Administration (FDA) for the discussions about the clinical applications, about SA and UQ, for their reviews on the draft, and the learning material they provided. Also, the authors want to acknowledge the crucial support of Adam J. Stephens from Sandia National Laborarories with the Dakota inputs.

## Author Contributions

**Conceptualization:** Alfonso Santiago, Richard A. Gray, Karen May-Newman, Pras Pathmanathan.

**Data curation:** Alfonso Santiago, Constantine Butakoff, Beatriz Eguzkitza, Karen May-Newman, Vi Vu.

**Formal analysis:** Alfonso Santiago, Beatriz Eguzkitza, Richard A. Gray, Pras Pathmanathan.

**Funding acquisition:** Richard A. Gray, Karen May-Newman, Mariano Vázquez.

**Investigation:** Alfonso Santiago, Richard A. Gray, Pras Pathmanathan.

**Methodology:** Alfonso Santiago, Richard A. Gray, Karen May-Newman, Pras Pathmanathan, Mariano Vázquez.

**Project administration:** Richard A. Gray, Karen May-Newman, Mariano Vázquez.

**Software:** Alfonso Santiago, Beatriz Eguzkitza, Mariano Vázquez.

**Supervision:** Richard A. Gray, Karen May-Newman, Pras Pathmanathan, Mariano Vázquez.

**Validation:** Alfonso Santiago, Richard A. Gray, Karen May-Newman, Pras Pathmanathan.

**Visualization:** Alfonso Santiago.

**Writing – original draft:** Alfonso Santiago, Constantine Butakoff, Richard A. Gray, Karen May-Newman, Pras Pathmanathan, Mariano Vázquez.

**Writing – review & editing:** Alfonso Santiago, Constantine Butakoff, Richard A. Gray, Karen May-Newman, Pras Pathmanathan, Mariano Vázquez.

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
