## [Decision Letter · Decision Letter 0]

12 Jan 2022

Dear Mr. Santiago,

Thank you very much for submitting your manuscript "Design and execution of a Verification, Validation, and Uncertainty Quantification plan for a numerical model of left ventricular flow after LVAD implantation" for consideration at PLOS Computational Biology.

As with all papers reviewed by the journal, your manuscript was reviewed by members of the editorial board and by two independent reviewers. In light of the reviews (below this email), we would like to invite the resubmission of a significantly-revised version that takes into account the reviewers' comments.

We cannot make any decision about publication until we have seen the revised manuscript and your response to the reviewers' comments. Your revised manuscript is also likely to be sent to reviewers for further evaluation.

Sincerely,

Andrew D. McCulloch, Ph.D.

Associate Editor

PLOS Computational Biology

Daniel Beard

Deputy Editor

PLOS Computational Biology

Reviewer's Responses to Questions

**Comments to the Authors:**

Reviewer #1: Uploaded as attachment

Reviewer #2: The authors presented a very through approach for the verification, validation and uncertainty quantification of a model of the LV-LVAD system. It is a first step towards the possible use of such simulations for in silico clinical trials.

However, even tough the study is overall complete and well designed, I would like to highlight a few points that would require further clarifications.

General comments

Could the authors please use units that are more commonly used in clinical practice (cardiac output in L/min and pressure in mmHg)?

Methods:

2.1. Could the authors please comment on the adequacy between the HM2 inflow cannula and the tygon tubing they used in the experiments?

2.1. The authors have retrieved the H-Q curves for each pump speed. Was it done for a “steady” state? Could the authors provide more information about the protocol they implemented to get these curves? They are the standard for LVADs but a few studies have also observed that the LVAD response is more complex and unsteady, which might explain some discrepancies between the model and the experiments.

2.2.1. Could the authors give more details about the solid mesh they used for the one-way FSI? (shell elements or solid elements?) What mechanical properties have they used for the solid model? (only the young modulus is given in the description of the experiments). Have the authors compared the overall mechanics of the solid domain between their model and the experiments?

2.2.3. Could the authors explain their choice to keep the mitral valve inlet at a constant pressure? To my understanding, this is not the case for the experiments. Overall, a sketch of the FSI model with the different BC would really help us understand the simulation.

2.2.4. The authors used a porous layer to model the valve, which is numerically more efficient. Have they assessed the effect of this model assumption on their results? In the experiments, the valve motion may impact the intraventricular flow patterns, especially the opening of the aortic valve.

2.2.5. The authors have approximated the pressure-flow relationship with a quadratic equation. I assumed the coefficients AVAD, BVAD and CVAD used in equation 3 are the fitting parameters and they depend on the LVAD speed. The authors provide the values in Table 5, but it is a bit late in the manuscript. Could they please add this table to the methods section?

Is this model commonly used for CFD models of LVAD?

2.2.5. Also, have the authors evaluated the cycle do cycle dependence of their model?

2.4.2. The sensitivity analysis is an important part of this study and would benefit for further details. Could the authors comment on the relevance of the Pearson’s coefficient for their approach? Their problem is quite complex and a linear assumption, without any interactions between the variables, might be a strong hypothesis. As they state, the second approach seems more appropriate.

Also, could the authors provide the equations for the Sobol indices they compute? as they present them later in the results sections. Could they also comment on their choice of a 5th order polynomial chaos?

Finally, they stated that both analyses relied on 500 samples. How were these samples chosen (i.e. which experimental plan did the authors used? Latin hypercube sampling?)?

2.4.3. The authors added a 10% error range in the Qols measured in the experiment. Have the authors measured the repeatability and reproducibility of the experiments? If not, was this 10% error arbitrary?

Results:

3. The part of the sampling of the input values using latin hyper cube sampling belong to the methods section. (it can be repeated in the results section but needs to appear in the methods).

3.1.1. Could the authors clarify their choice for the value ranges used for the SA? Was it based on previous studies or their experiments?

3.1.1. The second approach for the SA considered nonlinear effects and interactions between the parameters. However, the authors only present the coefficient of the linear effects. If the other Sobol coefficients are not significant or very low, the authors could state it clearly in the results section.

3.1.2. The authors have assumed that the H-Q curves were quadratic, defined with 3 parameters, but then they only use 2 parameters in the uncertainty characterization? Was the third parameter irrelevant or very small compared to the other ones?

Figure 4. The authors previously stated that the 4 Qols were the max and average aortic and LVAD flows. What is QRAT that the authors present in this figure?

3.2 The authors have performed the UQ analysis for 6 experimental points. They describe them later, but could they please add a short sentence to justify this number in this section?

Figure 6. Could the authors add units to all the plots? Also, the Qao plot is cut. Could the authors explain why they observe some flow experimentally (despite being minimal) through the LVAD, even is the RPM is 0?

3.2.4. The authors obtained the best agreement for the 8k RPM condition, especially for the flowrate through the LVAD. However, the flowrate in the aorta is more difficult to predict, due to the complex opening and closing of the aortic valve. Could the authors comment on the capacity of their valve model to predict such phenomena correctly? Could it partly explain the difference they observed (especially in the 0K RPM condition?)?

Conclusion:

5. The authors state that the bench experiments provide a highly reproducible set of comparators. Have they evaluated the reproducibility of the experiments? I assume they meant this comparatively to in vivo measurements.

Minor comments:

2.1. Please replace “during systole the ventricle contracts” by “it is contracted or compressed”. This sentence may lead people to think that there is some type of active material able to contract, when the piston pump is responsible for the contraction.

2.4.3. Please add a space between “metric.” and “To evaluate”. Please remove a space before the coma in the same sentence.

Table 3. Ejection fraction is either 0.35 or 35%.

Figure 5. The legend of the color bar is a little difficult to read.

3.3. Validation credibility factors. Please add “on” after “model is based”.

Figure 6. The difference in scale in the graphs of QVAD is large and tend to be misleading as it shows large difference when they are physiologically not important. The authors could change the scale of the y axis on these plots.

**Have the authors made all data and (if applicable) computational code underlying the findings in their manuscript fully available?**

Reviewer #1: **No: **See comments to the authors

Reviewer #2: Yes

PLOS authors have the option to publish the peer review history of their article (what does this mean?). If published, this will include your full peer review and any attached files.

Reviewer #1: **Yes: **Ahmet Erdemir

Reviewer #2: No
---

## [Decision Letter · Decision Letter 1]

29 Mar 2022

Dear Mr. Santiago,

Thank you very much for submitting your manuscript "Design and execution of a Verification, Validation, and Uncertainty Quantification plan for a numerical model of left ventricular flow after LVAD implantation" for consideration at PLOS Computational Biology. As with all papers reviewed by the journal, your revised manuscript was reviewed by members of the editorial board and by the original two independent reviewers. The reviewers noted some minor but important issues. Based on the reviews, we are likely to accept this manuscript for publication, providing that you revised the manuscript according to the review recommendations.

Sincerely,

Andrew D. McCulloch, Ph.D.

Associate Editor

PLOS Computational Biology

Daniel Beard

Deputy Editor

PLOS Computational Biology

[LINK]

Reviewer's Responses to Questions

**Comments to the Authors:**

Reviewer #1: The review is uploaded as an attachment.

Reviewer #2: Please see the attached document

**Have the authors made all data and (if applicable) computational code underlying the findings in their manuscript fully available?**

Reviewer #1: **No: **Experimental data, in particular time history of signals, used as simulation inputs and as output comparators, were not provided.

Reviewer #2: Yes

PLOS authors have the option to publish the peer review history of their article (what does this mean?). If published, this will include your full peer review and any attached files.

Reviewer #1: **Yes: **Ahmet Erdemir

Reviewer #2: No

Figure Files:

Data Requirements:

Reproducibility:

References:

---

## [Editor Report · Decision Letter 2]

26 Apr 2022

Dear Mr. Santiago,

We are pleased to inform you that your manuscript 'Design and execution of a Verification, Validation, and Uncertainty Quantification plan for a numerical model of left ventricular flow after LVAD implantation' has been provisionally accepted for publication in PLOS Computational Biology.

Best regards,

Andrew D. McCulloch, Ph.D.

Associate Editor

PLOS Computational Biology

Daniel Beard

Deputy Editor

PLOS Computational Biology

---

## [Editor Report · Acceptance letter]

3 Jun 2022

PCOMPBIOL-D-21-02033R2 

Design and execution of a Verification, Validation, and Uncertainty Quantification plan for a numerical model of left ventricular flow after LVAD implantation

Dear Dr Vazquez,

I am pleased to inform you that your manuscript has been formally accepted for publication in PLOS Computational Biology. Your manuscript is now with our production department and you will be notified of the publication date in due course.

With kind regards,

Agnes Pap
